# GENERATIVE BANDIT OPTIMIZATION VIA DIFFUSION POSTERIOR SAMPLING

## ABSTRACT

Many real-world discovery problems, including drug and material design, can be modeled within the bandit optimization framework, where an agent selects a sequence of experiments to efficiently optimize an unknown reward function. However, classic bandit algorithms operate on fixed finite or continuous action sets, making discovering novel designs impossible in the former case, and often leading to the curse of dimensionality in the latter, thus rendering these methods impractical. In this work, we first formalize the *generative bandit* setting, where an agent wishes to maximize an unknown reward function over the support of a data distribution, often called *data manifold*, which implicitly encodes complex constraints (e.g., the geometry of valid molecules), and from which (unlabeled) sample data is available (e.g., a dataset of valid molecules). We then propose **D**iffusion **P**osterior **S**ampling (DIFFPS), an algorithm that tackles the exploration-exploitation problem directly on the learned data manifold by leveraging a conditional diffusion model. We formally show that the statistical complexity of DIFFPS adapts to the *intrinsic dimensionality* of the data, overcoming the curse of dimensionality in high-dimensional settings. Our experimental evaluation supports the theoretical claims and demonstrates promising performance in practice.

## 1 INTRODUCTION

Many real-world discovery problems, spanning drug discovery (Schneider, 2018), material design (Guo et al., 2021), and circuit design (El-Turky & Perry, 1989) among others, can be framed as bandit optimization (Lattimore & Szepesvári, 2020). In this context, an agent aims to optimize an unknown (black-box) reward function $r$ over an experiments space $\Omega$. Crucially, since $r$ is unknown, and evaluating $r(x)$ for $x \in \Omega$ is typically expensive, the agent needs to select wisely a sequence of experiments $x_1, \ldots, x_T$ that balances efficient exploration to learn $r$, and exploitation of its current belief to select promising maximizers, a challenge known as the *exploration-exploitation dilemma*. Historically, bandit algorithms were first devised for fixed and finite action sets, where the agent is given a set $\Omega = \{x_1, \ldots, x_A\}$, which does not allow discovering novel actions (e.g., molecules, previously unknown to the algorithm designer). More recently, bandit optimization algorithms have been extended to continuous action spaces (Srinivas et al., 2009; Abbasi-Yadkori et al., 2011), e.g., $\Omega = \mathbb{R}^D$, where decision-making occurs in a known or learned $D$-dimensional data representation space. Unfortunately, for many real-world problems, including most scientific discovery applications, the *ambient dimensionality* $D$ is very high, causing bandit algorithms to incur statistical complexities too large to be practical (Djolonga et al., 2013; Kandasamy et al., 2015). In other words, these algorithms suffer the *curse of dimensionality* as their practical and theoretical sample complexities, i.e., number of experiments needed to discover maximizers, heavily depend on $D$. Moreover, in most real-world problems, such as molecular design, most points (or actions) in $\Omega = \mathbb{R}^D$ do not correspond to valid molecules. Thus, fixed finite action spaces are too restrictive for discovery or too large to enumerate, while typical continuous spaces lead to the curse of dimensionality and cannot easily distinguish between valid experiments and invalid ones, e.g., an invalid molecule.

To address this issue, we introduce the *generative bandit* setting, aiming to close the gap between finite and continuous action sets by combining their advantages: the ability to discover valid actions unknown a priori to the algorithm designer, while tackling the curse of dimensionality in high-dimensional real-world problems (Sec. 3). While previous works attempt to solve the bandit problem

on a learned low-dimensional latent space (Gómez-Bombarelli et al., 2018; Grosnit et al., 2021), in generative bandits the action space is unknown to the agent and is defined as the support of a possibly complex data distribution $P_x$ approximately learnable through sample data, e.g., a dataset of known molecules. This set, namely $\Omega = \mathrm{supp}(P_x)$, typically called *data manifold*, can capture implicit constraints hidden in the data, e.g., the complex geometry of valid molecules, and its dimensionality is denoted as *intrinsic data dimensionality* (Fefferman et al., 2016). According to the *manifold hypothesis*, the intrinsic dimensionality $m$ of $\Omega$ is significantly lower than the ambient dimensionality, i.e., $m \ll D$, for a wide range of real-world data types (Fefferman et al., 2016; Stanczuk et al., 2024). As a consequence, in this work we first aim to answer the following question:

*How can a decision-making agent solve the exploration-exploitation problem directly on the unknown data manifold?*

To this end, and motivated by the success of diffusion models (Sohl-Dickstein et al., 2015; Song & Ermon, 2019; Ho et al., 2020) in learning complex data distributions across various domains, including chemistry (Hoogeboom et al., 2022), biology (Corso et al., 2022), and robotics (Chi et al., 2023), we present **D**iffusion **P**osterior **S**ampling (DIFFPS), which extends classic posterior sampling (Russo & Van Roy, 2014; Osband & Van Roy, 2017) to generate a sequence of approximately valid actions from diverse areas of the unknown manifold via sequential conditional generation, gradually concentrating the generated experiments on high-reward regions (Sec. 4).

Next, by leveraging recent theoretical results on provable manifold learning via diffusion (Chen et al., 2023; Stanczuk et al., 2024), we shed light on the statistical complexity of DIFFPS, showing that under certain structural assumptions, it adapts to the intrinsic data dimensionality $m$, thus overcoming the curse of dimensionality that typically hinders the applicability of bandit algorithms in real-world discovery problems (Hao et al., 2020; Djolonga et al., 2013; Kandasamy et al., 2015) (Sec. 5). Finally, we provide an experimental evaluation of DIFFPS, supporting our theoretical claims empirically and showing promising performance (Sec. 6).

To sum up, we make the following contributions:

- The *generative bandit* setting, where the action set $\Omega$ is the unknown support, also called *data manifold*, of a complex data distribution $P_x$ learnable from unlabeled data (Sec. 3).
- **D**iffusion **P**osterior **S**ampling (DIFFPS), an algorithm that leverages conditional diffusion models to tackle the exploration-exploitation problem directly on the learned data manifold, and **G**enerative **P**osterior **S**ampling (GENPS), a generative model agnostic generalization of DIFFPS (Sec. 4).
- A statistical analysis of the (Bayesian) regret incurred by DIFFPS, showing that it adapts to the *intrinsic data dimensionality*, and an analysis of the *misgeneration* regret of DIFFPS (Sec. 5).
- An experimental evaluation of DIFFPS, providing empirical support for our theoretical claims and demonstrating promising performance. (Sec. 6).

## 2 BACKGROUND AND NOTATION

We denote with $[N]$ a set of integers $\{1, \ldots, N\}$. Let $X$ be a set, $\Delta(X)$ is the probability simplex over $X$. Given a probability distribution $P \in \mathcal{P}(\mathbb{R}^D)$, we indicate with $\mathrm{supp}(P) \coloneqq \{x \in \mathbb{R}^D : P(x) > 0\}$ the support of $P$.

### 2.1 BANDIT OPTIMIZATION, EXPLORATION-EXPLOITATION, AND POSTERIOR SAMPLING

**Bandit optimization.** A $T$-round bandit (optimization) problem (Lattimore & Szepesvári, 2020) is a tuple $\upsilon = \langle \Omega, r_{\theta_*}, T \rangle$, where $\Omega \subseteq \mathbb{R}^D$ is a (possibly infinite) set of actions, $r_{\theta_*} : \Omega \to \mathbb{R}$ is an unknown deterministic reward function, and $T$ is the number of rounds. At every round $t \in [T]$, an agent selects an action $x_t \in \Omega$ according to a policy $\pi = \{\pi_t\}_{t \in [T]}$ with $\pi_t \in \mathcal{P}(\mathbb{R}^D)$, and receives the noisy feedback $y_t = r_{\theta_*}(x_t) + \epsilon_t$, i.e., the reward function evaluated at $x_t$ plus zero-mean noise.

**Exploration-exploitation problem and posterior sampling.** Balancing exploration of novel actions to learn $r_{\theta_*}$, and exploitation of the current belief about $r_{\theta_*}$ to propose promising actions, is known as the exploration-exploitation dilemma. A classic algorithm to address this challenge is posterior sampling (PS) (Russo & Van Roy, 2014). Given a set of bandit instances $\{\upsilon = \langle \Omega, r_\theta, T \rangle\}_{\theta \in \Theta}$ and a prior distribution $q_1$ over $\Theta$, PS operates as follows. At each round $t \in [T]$, the agent samples a

reward parameter $\tilde{\theta}_t \sim q_t$, computes the policy $\pi_t$ that maximizes $r_{\tilde{\theta}_t}$, selects an action $x_t \sim \pi_t$, receiving a noisy feedback $r_{\theta_*}(x_t) + \epsilon_t$ from the true reward model. The agent then updates the posterior $q_{t+1}$ to integrate the new evidence. By acting optimally with respect to sampled reward functions (thus promoting exploration) and updating its beliefs based on observed feedback, the agent gradually learns enough about the true reward function to eventually act optimally with respect to it.

## 2.2 Diffusion models, score matching, and conditional generation

**Generative models and conditional generation.** Given i.i.d. samples from an unknown data distribution $P_x$, generative models aim to learn an approximate distribution $\widehat{P}_x$ that closely matches $P_x$. For a joint distribution $P_{xy}$, where $y$ is a label for sample $x$, we express the conditional distribution as $P(\cdot \mid y)$ and its learned approximation as $\widehat{P}(\cdot \mid y)$. For the sake of clarity, in the following we denote as $P = P_x$ the generative model exactly capturing the data distribution.

**Conditional diffusion models and score matching with neural networks.** Given a random variable $x^0 \sim P_x$ diffusion models (DMs) construct a sequence of random variables $x^0, x^1, \ldots, x^K$ by sequentially adding Gaussian noise (Song et al., 2020). This *forward process* transforms the data distribution into a noise distribution. DMs learn the *backward process* to convert noise back into the original data distribution. In conditional diffusion models, we aim to sample from $P(\cdot \mid y)$ rather than $P_x$. The noising process can be expressed via the following forward Ornstein–Uhlenbeck SDE:

$$\mathrm{d}x(k) = -\frac{1}{2}g(k)x(k)\mathrm{d}k + \sqrt{g(k)}\mathrm{d}w(k) \quad k \in (0, K] \tag{1}$$

where $x(0) \sim P^0(\cdot \mid y)$, $K$ is the terminal time, $w$ is a Wiener process, and the initial distribution $P^0(\cdot \mid y)$ is induced by $P_{xy}$. For clarity, we set $g(k) = 1$. We denote with $P^k(\cdot \mid y)$ the distribution of $x(k)$ and with $p^k(x \mid y)$ its density. We define the conditional score at time $k$ as $\nabla_x \log p^k(x \mid y)$, which in principle can be estimated by solving the following minimization problem:

$$\arg\min_{s \in \mathcal{S}} \mathbb{E}_{k \sim \mathcal{U}(k_0, K)} \mathbb{E}_{(x,y) \sim P^k} \left[ \|\nabla_x \log p^k(x \mid y) - s(x, y, k)\|_2^2 \right] \tag{2}$$

where $\mathcal{S}$ is a properly defined concept class and $\mathcal{U}$ denotes the uniform distribution (Song et al., 2020). Unfortunately, this problem is intractable as $\nabla_x \log p^k(x \mid y)$ is unknown. However, the same solution can be obtained by minimizing over $s \in \mathcal{S}$ the following loss function, as in (Li et al., 2024):

$$\mathbb{E}_{(x,y) \sim P_{xy}} \ell(x, y, s) = \mathbb{E}_{(x,y) \sim P_{xy}} \mathbb{E}_{k \sim \mathcal{U}(k_0, K)} \mathbb{E}_{x' \sim \mathcal{N}(\alpha(k)x, h(k)I_D)} \left[ \|\nabla_{x'} \log \phi^k(x' \mid x) - s(x', y, k)\|_2^2 \right]$$

Hereby, $\phi^k(x' \mid x)$ is the density of $\mathcal{N}(\alpha(k)x, h(k)I_D)$, the conditional distribution of $x(k)$ given $x(0)$ with $\alpha(k) := \exp(-k/2)$ and $h(k) := 1 - \exp(-k)$. In the following, we denote with $\hat{s}$ the score obtained by solving the above problem approximately by estimating the expectations with data.

**Conditional generation via diffusion.** Once an estimate $\hat{s}$ for the conditional score function is available, new samples can be obtained by simulating the following reverse-time SDE:

$$\mathrm{d}x(k) = \left[ \frac{1}{2}x(k) + \hat{s}(x(k), y, k) \right] \mathrm{d}k + \mathrm{d}\bar{w}(k) \tag{3}$$

where $x(K) \sim \mathcal{N}(0, I_D)$ and $\bar{w}$ is a reversed Wiener process.

## 3 Problem Setting: Generative Bandits with Offline Data

In this section, we first introduce the *generative bandit* problem, extending bandit optimization to settings where the valid action set $\Omega$ is the unknown support of a (typically complex) data distribution, often regarded as *data manifold*[1]. Then, along with the classic Bayesian regret (Lattimore & Szepesvári, 2020), we introduce a performance measure named *misgeneration regret*, which captures the cost due to generating invalid actions, i.e., $x_t \notin \Omega$, resembling measures of constraint violation in bandit or reinforcement learning with safety constraints (Amani et al., 2019; Efroni et al., 2020).

---

[1]Here the term manifold is used in a loose sense. Specific structure, e.g., compactness (Stanczuk et al., 2024), linearity (Chen et al., 2023), is typically assumed to derive theoretical results, as later done in Sec. 5.

### 3.1 ONLINE LEARNING INTERACTION PROCESS

**Definition 1** (Generative Bandit). *A $T$-round generative bandit (optimization) problem is a tuple $\upsilon = \langle P_x, r_{\theta_*}, c, T \rangle$, where $r_{\theta_*}$, also expressed as $r_*$, and $T$ denote respectively an unknown reward function and the interaction budget. The action set corresponds to the data manifold and is implicitly defined as $\Omega := \mathrm{supp}(P_x)$, where $P_x$ is an unknown data distribution. $c : \mathbb{R}^D \to \mathbb{R}$ is an unknown validity function assigning positive penalty to invalid actions $x \notin \Omega$, while $c(x) = 0$ for $x \in \Omega$.*

The interaction process proceeds as follows: at every round $t \in [T]$, the agent selects an action $x_t \in \mathbb{R}^D$ (also referred to as *experiment* or *design*) according to a policy $\pi := \{\pi_t\}_{t \in [T]}$ where $\pi_t \in \mathcal{P}(\mathbb{R}^D)$ (i.e., $x_t \sim \pi_t$), and receives a noisy observation $y_t = r_*(x_t) + \epsilon_t$, with $\epsilon_t$ being conditionally $R$-sub-Gaussian noise (Vershynin, 2018). If action $x_t$ is invalid (i.e., $x_t \notin \Omega$), the agent incurs an unobserved penalty $c(x_t)$. Here, we consider the case where the agent cannot query the validity function $c$, while in Sec. 6, we discuss how black-box access to $c$ can improve performance.

**Access to offline unlabeled data**  To solve a generative bandit problem, an agent must learn to distinguish valid actions ($x \in \Omega$) from invalid ones ($x \notin \Omega$). To this end, and to capture practical settings, we assume the agent has access to an unlabeled dataset $\mathcal{D}_{\mathrm{unlabeled}} := \{(x_i)\}_{i=1}^n$ composed of $n$ i.i.d. unlabeled points sampled from the unknown data distribution $P_x$, namely $x_i \sim P_x, \;\; \forall\, i \in [n]$.

### 3.2 OPTIMALITY MEASURES: BAYESIAN REWARD AND MISGENERATION REGRET

We now introduce performance measures to account for both the cost of proposing sub-optimal actions w.r.t. the unknown true reward $r_*$, and the penalty due to playing invalid actions (i.e., $x_t \notin \Omega$).

**Definition 2** (Bayesian reward and misgeneration regret). *Given a set of generative bandit instances $\{\upsilon = \langle P_x, r_\theta, c, T \rangle\}_{\theta \in \Theta}$ with prior $q$ over $\Theta$, we define the Bayesian reward and misgeneration regret incurred by a policy $\pi = \{\pi_t\}_{t \in [T]}$ as follows:*

$$\mathcal{BR}_r(T, \pi) := \mathbb{E}_{\theta_* \sim q} \left[ \sum_{t=1}^{T} r_*(x^*) - \mathbb{E}_{x_t \sim \pi_t} [r_*(x_t)] \right] \qquad \text{(reward regret)}$$

$$\mathcal{BR}_c(T, \pi) := \mathbb{E}_{\theta_* \sim q} \left[ \sum_{t=1}^{T} \mathbb{E}_{x_t \sim \pi_t} [c(x_t)] \right] \qquad \text{(misgeneration regret)}$$

*where we use $r_*$ to denote $r_{\theta_*}$, and define $x^* \in \arg\max_{x \in \Omega} r_*(x)$.*

The term $\mathcal{BR}_r(T, \pi)$ represents the expected regret over the instance class $\Theta$ incurred by the agent from proposing sub-optimal actions w.r.t. the unknown reward function $r_*$. Conversely, $\mathcal{BR}_c(T, \pi)$ quantifies the expected regret over $\Theta$ due to proposing invalid samples (i.e., $x_t \notin \Omega$), e.g., invalid molecules, measured via the validity function $c$ in Definition 1.

Intuitively, a policy minimizing the reward and misgeneration regret measures in Definition 2 must use the interaction budget $T$ wisely to efficiently balance exploration and exploitation within the (potentially complex) support of the unknown data distribution $P_x$, i.e., the data manifold $\Omega := \mathrm{supp}(P_x)$. In the next section, we propose an algorithm that tackles this challenging problem by sequential conditional generation via diffusion modeling (Song & Ermon, 2019; Ho et al., 2020).

## 4 DIFFUSION POSTERIOR SAMPLING WITH OFFLINE UNLABELED DATA

In the following, we present **D**iffusion **P**osterior **S**ampling (DIFFPS), an algorithm that leverages diffusion models (Song & Ermon, 2019) to tackle the generative bandit problem, as in Definition 1.

At each iteration $t \in [T]$, DIFFPS (see Algorithm 1) uses a conditional diffusion model to generate an action $x_t \sim \hat{\pi}_t$ from the region of the manifold $\widehat{\Omega}_{\tilde{r}_t} \approx \Omega_{\tilde{r}_t} := \{x \in \Omega : x \in \arg\max_{x \in \Omega} \tilde{r}_t(x)\} \subseteq \Omega$ of approximately valid actions maximizing the imaginary reward function $\tilde{r}_t$ sampled from the reward prior $q_t$. As illustrated in Fig. 1, this process enables DIFFPS to sequentially (and approximately) explore different regions $\{\Omega_{\tilde{r}_t}\}_{t \in [T]}$ of the unknown manifold, and by integrating observations into

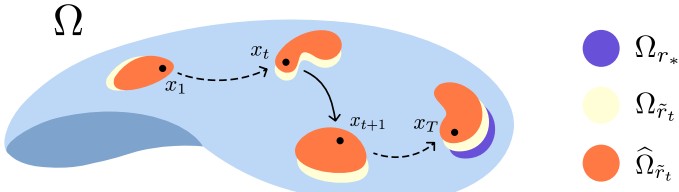

Figure 1: Data manifold $\Omega = \text{supp}(P)$. In yellow: manifold regions $\{\Omega_{\tilde{r}_t}\}_{t \in [T]}$ of actions maximizing imaginary rewards $\{\tilde{r}_t\}_{t \in [T]}$. In orange: approximate regions used for sampling, e.g., $\widehat{\Omega}_r \approx \Omega_r$. In purple: region $\Omega_{r_*}$ of maximizers of true reward function $r_*$.

the reward prior $q_t$ gradually learn the true reward function $r_*$ well enough to ultimately approximately sample from the region $\Omega_{r_*} \subseteq \Omega$ of valid actions maximizing the true unknown reward function $r_*$.

---

**Algorithm 1** DIFFPS: Diffusion Posterior Sampling (with offline unlabeled data)

1: **Input:** $T$ : number of online samples, $q_1$ : reward parameter prior, $\mathcal{D}_{\text{unlabeled}}$ : $n$ unlabeled data, $k_0$ : early-stopping time, $\nu$ : noise level
2: **for** $t = 1, 2, \ldots, T$ **do**
3:      Sample reward parameter $\tilde{\theta}_t \sim q_t$ and define $\tilde{r}_t := r_{\tilde{\theta}_t}$
4:      Label data in $\mathcal{D}_{\text{unlabeled}}$ via $\tilde{r}_t$: $\mathcal{D} := \{(x^i, y^i := \tilde{r}_t(x^i) + \xi_i)\}_{i=1}^n$ with $\xi_i \sim \mathcal{N}(0, \nu^2)$
5:      Conditional score matching on dataset $\mathcal{D}$ and arbitrary function class $\mathcal{S}$:

$$\hat{s} \in \arg\min_{s \in \mathcal{S}} \mathop{\mathbb{E}}_{(x,y) \in \mathcal{D}} \mathop{\mathbb{E}}_{k \sim \mathcal{U}(k_0, K)} \mathop{\mathbb{E}}_{x' \sim \mathcal{N}(\alpha(k)x, h(k)I_D)} \left[ ||\nabla_{x'} \log \phi^k(x' \mid x) - s(x', y, k)||_2^2 \right]$$

6:      Compute maximum imaginary reward: $\tilde{y}_t = \max_{x \in \Omega} \tilde{r}_t(x)$
7:      Sample action $x_t := x_t(0)$ via reverse SDE induced by estimated conditional score $\hat{s}_t(\cdot, \tilde{y}_t, \cdot)$:

$$\mathrm{d}x(k) = \left[ \frac{1}{2} x(k) + \hat{s}(x(k), \tilde{y}_t, k) \right] \mathrm{d}k + \mathrm{d}\bar{w}(k)$$

8:      Play $x_t$ and observe $y_t = r_*(x_t) + \epsilon_t$
9:      Compute $q_{t+1}$ via posterior update as in Eq. 6
10: **end for**

---

In the following, we present a detailed explanation of Algorithm 1. First, at each iteration $t \in [T]$, DIFFPS samples an imaginary reward parameter from the rewards prior, namely $\tilde{\theta}_t \sim q_t$ (line 3). Then, it computes the labeled dataset $\mathcal{D}$ via labeling the dataset $\mathcal{D}_{\text{unlabeled}}$ by defining pairs $(x^i, y^i)$ with $y^i := \tilde{r}_t(x^i) + \xi_i$, where we define $\tilde{r}_t := r_{\tilde{\theta}_t}$ and $\xi_i \sim \mathcal{N}(0, \nu^2)$ (line 4). Afterwards, DIFFPS learns a conditional diffusion model $\widehat{P}_t(\cdot \mid y)$ by estimating the score $\hat{s}$ via conditional score matching on dataset $\mathcal{D}$ (line 5), and computes the maximum imaginary reward value over $\Omega$, namely $\tilde{y}_t$ (line 6). Once $\tilde{y}_t$ is computed, the algorithm approximately samples via conditional generation $x_t \sim \hat{\pi}_t = \widehat{P}_t(\cdot \mid \tilde{y}_t)$ from the region of the manifold achieving reward $\tilde{y}_t$, namely $\Omega_{\tilde{r}_t}$ (line 7). Ultimately, it plays action $x_t$ to observe feedback $r_*(x_t) + \epsilon_t$ (line 8), and performs posterior update of the reward prior $q_t$ (line 9) to integrate the new evidence gained about the true reward function $r_*$.

**Towards a practical and scalable algorithm.** The oracle optimization step (line 6) is a maximization problem over $\Omega$. We approximate this using output-space optimization techniques leveraging the generative model $\widehat{P}$, supported on the approximate data manifold $\widehat{\Omega}$, as by Krishnamoorthy et al. (2023). In Apx. F, we present two alternative oracle implementations, which can optionally exploit black-box access to the validity function $c$ to improve performances as discussed in Sec. 6.

Moreover, it is not necessary to retrain the diffusion model at each iteration $t$ as one can leverage the score decomposition $\nabla_x \log p(x|y) = \nabla_x \log p(x) + \nabla_x \log p(y|x)$, train a score model for $p(x)$ on the unlabeled dataset, and use $\tilde{r}_t$ for guidance (Song et al., 2020). Although tackling scalable uncertainty quantification is beyond the scope of this work, recent approximate posterior

sampling methods (Osband et al., 2023) that have shown promising performances for exploration in LLMs (Dwaracherla et al., 2024) can straightforwardly be integrated with DIFFPS.

**Exploration-exploitation directly on the learned data manifold.** Crucially, by generating actions via conditional sampling DIFFPS effectively explores only the learned manifold $\widehat{\Omega} \approx \Omega$ using a learned sampler (i.e., the diffusion process), without relying on an explicit representation of the action space $\Omega$. Formally, one can see that for all $t \in [T]$, action $x_t$ is sampled approximately in-manifold:

$$x_t \sim \hat{\pi}_t = \widehat{P}_t(\cdot \mid \tilde{y}_t) \text{ and } \widehat{\Omega}_{\tilde{r}_t} \coloneqq \operatorname{supp}\left(\widehat{P}_t(\cdot \mid \tilde{y}_t)\right) \subseteq \operatorname{supp}(\widehat{P}) \eqqcolon \widehat{\Omega} \approx \Omega \tag{4}$$

Here, $\widehat{P}$ stands for the unconditional generative model trained on the unlabeled data $\mathcal{D}_{\text{unlabeled}}$ following distribution $P_x$. Interestingly, this logic does not rely on the specific structure of diffusion models, and in Apx. B, we present a generative model agnostic generalization of Algorithm 1.

Intuitively, solving the exploration-exploitation problem within the learned data manifold rather than in the entire ambient space might significantly reduce the number of samples needed to discover maximizers of the unknown reward function. In the next section, we formally prove this intuition under typical structural assumptions, showing that the statistical complexity of DIFFPS adapts to the intrinsic dimensionality of the data manifold.

# 5 THEORETICAL GUARANTEES: REWARD AND MISGENERATION REGRET

In this section, we present an upper bound on the Bayesian reward and misgeneration regrets, as in Definition 2, achieved by DIFFPS against an optimal sampling strategy. This result captures the impact on statistical complexity of solving the exploration-exploitation problem directly on the learned data manifold. This gain can be formally captured via the notion of intrinsic data dimensionality[2].

> **Definition 3** (Intrinsic data (manifold) dimensionality). *Given a data distribution $P_x$ with support $\Omega \coloneqq \operatorname{supp}(P_x)$, we define:*
> $$m(\Omega) \coloneqq \min\{m \in \mathbb{N} : \Omega \subseteq \mathbb{R}^m\}$$

This complexity measure, which we denote as $m$ when $\Omega$ is clear from context, is clearly data dependent as it varies for different data types, e.g., molecules, natural images, proteins. Moreover, the well-known *manifold hypothesis* states that the intrinsic data dimensionality $m$ is significantly smaller than the ambient dimensionality $D$, namely $m \ll D$, in a variety of real-world problems (Loaiza-Ganem et al., 2024; De Bortoli, 2022; Fefferman et al., 2016; Valdés & Tchagang, 2023). To leverage the intrinsic data dimensionality in our analysis, we first assume the following.

**Assumption 5.1** (Low-dimensional linear subspace). *The action set $\Omega \coloneqq \operatorname{supp}(P_x)$ lives in a $m$-dimensional linear subspace. Namely, there exists an unknown matrix $V \in \mathbb{R}^{D \times m}$ with orthonormal columns such that $x = Vz$, where $z \in \mathbb{R}^m$ is a latent variable, and $D$ is the ambient dimensionality.*

**Assumption 5.2** (Linear bounded rewards and actions). *We assume that $r_*(x) = \theta_*^\top (\Pi_V x) \in [0, 1]$, where $\Pi_V = VV^\top$ is a projection onto $\Omega$, $\|\theta_*\|_2 = 1$, and $\|x_t\|_2 \le L \ \forall t \in [T]$.*

As stated in Definition 2, we wish to analyse two types of regret: the reward regret $\mathcal{BR}_r(T, \hat{\pi})$, which captures the in-manifold reward sub-optimality due to policy learning and approximate sampling, and the *misgeneration regret* $\mathcal{BR}_c(T, \hat{\pi})$, that captures the cost associated with generating invalid designs, i.e., out-of-manifold, namely $x_t \notin \Omega$. We now proceed to bound these two terms separately. As a first step in this direction, we state the following decomposition result for the reward regret:

> **Proposition 1** (Bayesian reward regret decomposition). *Given a policy $\hat{\pi}$ corresponding to running Algorithm 2, we have:*
> $$\mathcal{BR}_r(T, \hat{\pi}) \le \underbrace{\sum_{t=1}^T \mathop{\mathbb{E}}_{\theta_* \sim q} \mathop{\mathbb{E}}_{x_t \sim \pi_t} |r_*(x^*) - r_*(x_t)|}_{\mathcal{BR}_r^\Omega(T, \hat{\pi})} + \underbrace{\sum_{t=1}^T \mathop{\mathbb{E}}_{\theta_* \sim q} \left| \mathop{\mathbb{E}}_{x_t \sim \hat{\pi}_t} [r_*(x_t)] - \mathop{\mathbb{E}}_{x_t \sim \pi_t} [r_*(x_t)] \right|}_{\Delta_{(\Omega, \widehat{\Omega})}(T, \hat{\pi})}$$

---

[2]Notice that this definition is tight only for linear subspaces as later stated in Assumption 5.1.

Notice that this result, which is proved in Appendix D, is generative model agnostic and extends the result of Li et al. (2024, Appendix B.3.1) for conditional generation interpreted as offline bandit (Sakhi et al., 2023) to the (online) bandit setting. Crucially, Proposition 1 shows that the in-manifold reward sub-optimality incurred by policy $\hat{\pi}$ over $T$ interactions, decomposes into two terms: $\mathcal{BR}_r^\Omega(T, \hat{\pi})$ and $\Delta_{(\Omega, \widehat{\Omega})}(T, \hat{\pi})$. The former corresponds to the (Bayesian) regret of solving a classic bandit problem on the low-dimensional manifold by following the exact policy $\pi$, which does not account for the sampling approximation error. The latter accounts for the in-manifold reward sub-optimality caused by the gap between the exact policy $\pi$ and the approximate policy $\hat{\pi}$. This discrepancy arises because the quality of the learned conditional diffusion model is epistemically bounded by the amount $n$ of the available offline data in $\mathcal{D}_{\text{unlabeled}}$ and their data distribution $P_x$.

In the following, we will analyse the terms $\mathcal{BR}_r^\Omega(T, \hat{\pi})$ and $\Delta_{(\Omega, \widehat{\Omega})}(T, \hat{\pi})$ separately, bounding the former in a generative model agnostic way, and the latter by leveraging the specific diffusion model structure via recent statistical results for approximate conditional generation via diffusion models (Chen et al., 2023; Li et al., 2024). First, for the sake of analysis, we assume the following.

**Assumption 5.3** (Latent distribution and score realizability). *The latent variable $z$ follows distribution $\mathcal{N}(0, \Sigma)$ where $\lambda_{\min} I_m \preceq \Sigma \preceq \lambda_{\max} I_m$ with $\lambda_{\min} \leq \lambda_{\max} \leq 1$ and $\lambda_{\min} > 0$. Moreover, the true score is realizable, i.e., $\nabla_x \log p^k(x \mid y) \in \mathcal{S}$.*

As a design choice, we select the validity function to be $c(x) = \|(I_D - \Pi_V)x\|_2$, where $(I_D - \Pi_V)$ is the projection onto the orthogonal complement of $\Omega$. Therefore, for $x \in \Omega$ we have $c(x) = 0$.

Notice that Assumption 5.2 is typically made in the literature on high-dimensional bandits (e.g., (Lale et al., 2019)), while Assumptions 5.1 and 5.3 have been used to analyse diffusion models under the manifold hypothesis (e.g., Li et al., 2024; Chen et al., 2023). Moreover, for the sake of analysis, we consider the neural networks model class $\mathcal{S}$ with $m$-dimensional encoder-decoder structure to approximate the score function, as defined in (Li et al., 2024, Equation 4.8), and reported for completeness in Appendix E. We can finally state the following upper bounds.

**Theorem 5.1** (Bayesian reward and misgeneration regret upper bound). *Given a policy $\hat{\pi}$ corresponding to running Algorithm 1 and the assumptions stated above, by choosing $k_0 = ((Dm^2 + D^2m)/n)^{1/6}$, $\nu = 1/\sqrt{D}$, and $D \geq m^2$, defining $\bar{y} := \max_{t \in [T]} \tilde{y}_t$, we have:*

$$\mathcal{BR}_r(T, \hat{\pi}) = \widetilde{O}\left(m\sqrt{T} + T \cdot \text{OnlineDS}(T) \left(\frac{m^2 D + D^2 m}{n}\right)^{\frac{1}{6}} \cdot \bar{y}\right) \qquad \text{(reward regret)}$$

$$\mathcal{BR}_c(T, \hat{\pi}) = \widetilde{O}\left(T\left(\sqrt{k_0 D} + \sqrt{\frac{mD}{n^{1/2}}} \cdot \sqrt{\bar{y}^2 + m}\right)\right) \qquad \text{(misgeneration regret)}$$

*where $\text{OnlineDS}(T)$ is defined in Eq. 5.*

In the following, we briefly discuss the main insights from Theorem 5.1.

**Exploration-exploitation on the learned data manifold.** The (Bayesian) reward regret bound decomposes into two additive terms. The first matches the classic Bayesian regret for posterior sampling (with linear rewards) on an $m$-dimensional action space (Russo & Van Roy, 2014), thus DIFFPS approximately solves the exploration-exploitation problem on the low-dimensional learned manifold. Meanwhile, the second term captures the regret due to using the learned manifold as a misspecified action set (Freedman et al., 2021), showing a dependency on the *online distribution shift* defined as:

$$\text{OnlineDS}^2(T) := \max_{t \in [T]} \frac{\mathbb{E}_{P_{x|y=\tilde{y}_t}}[\ell(x, \tilde{y}_t; \hat{s})]}{\mathbb{E}_{P_{x,y}}[\ell(x, y; \hat{s})]} \tag{5}$$

Recalling that $\ell(x, y; \hat{s})$ represents the score estimation error at $(x, y)$, $\text{OnlineDS}(T)$ captures the worst-case ratio between the expected score error according to the exact policy $\pi_t = P(\cdot \mid y = \tilde{y}_t)$, and that under the joint distribution $P_{x,y}$. This joint distribution is determined by the offline data distribution $P_x$ and the imaginary reward model $\tilde{r}_t$ as $y = \tilde{r}_t(x) + \xi$. This term extends the distribution shift notion of Li et al. (2024) to the online setting, with the main difference that the numerator in Eq. 5 depends on the imaginary rewards $\tilde{r}_t$ computed by the algorithm, rather than on a value set a priori by the algorithm designer as typically the case with conditional generation. To sum up, OnlineDS captures the effect of the generative model quality on the reward regret of DIFFPS.

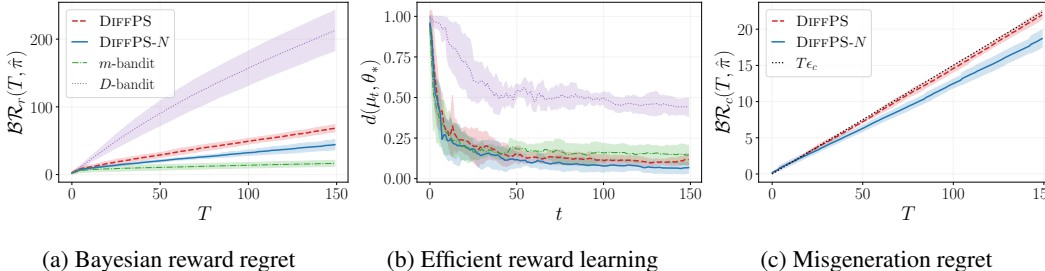

| (a) Bayesian reward regret | (b) Efficient reward learning | (c) Misgeneration regret |

Figure 2: Performance of DIFFPS and DIFFPS-$N$ against $m$-bandit and $D$-bandit baselines in terms of Bayesian reward regret (a) and reward learning (b) in a high-dimensional setting with unknown intrinsic data dimensionality $m$. In plot (c), it is shown the misgeneration regret for $\epsilon_c = 0.15$, controllable with DIFFPS-$N$ if black-box access to $c$ is available.

**No-(Bayesian) reward regret via increasing offline data $n$.** Since the action set $\Omega$ is unknown in generative bandits (see Definition 1), exploration-exploitation involves both the reward function $r_*$ and $\Omega$. Without online access to new data to refine $\Omega$, we learn the action manifold solely from offline data. Consequently, choosing $n = \widetilde{O}(T^3)$ renders the reward regret sub-linear in the experiment budget $T$ (Theorem 5.1). However, the misgeneration regret retains a sublinear dependence on the ambient dimensionality $\widetilde{O}(\sqrt{k_0 D})$. As explained in Sec. 6, this can be mitigated by querying the validity oracle $c(x_t)$ before evaluating the black-box reward $r_*$ on $x_t$.

In this section, we have shown that the statistical complexity of DIFFPS adapts to the intrinsic data dimensionality given certain assumptions. But does this behaviour happens also when some assumptions used for theoretical analysis (e.g., known intrinsic data dimensionality $m$) do not hold? In the following, we present an experimental evaluation of DIFFPS answering positively to this point.

## 6 EXPERIMENTAL EVALUATION

In this section, we perform an experimental evaluation of DIFFPS in a setting where the intrinsic data dimensionality $m$ is unknown to the algorithm, as opposed to Theorem 5.1 in Sec. 5. In particular, we aim to analyse the following aspects.

1. The Bayesian reward regret (see Definition 2) of DIFFPS (in Fig. 2a).
2. The ability of DIFFPS to perform efficient reward learning (in Fig. 2b).
3. The misgeneration regret (see Definition 2) and how it can be controlled when black-box access to the validity function $c$ is available (in Fig. 2c).

We consider a setting where $\Omega$ is a $m$-dimensional sphere embedded in $D$ dimensions. We set $D = 64$ and $m = 4$, consider a linear reward function with standard Gaussian prior $\theta_* \sim \mathcal{N}(0, I_D)$, and define $c(x)$ as the $l_2$ distance from the data manifold. In these experiments, DIFFPS knows neither $\Omega$ nor $m$. The oracle step (line 6 in Alg. 1) is implemented by selecting the maximum achieved within $\mathcal{D}$. While DIFFPS then samples a unique action, DIFFPS-$N$ samples $N$ actions and selects promising and approximately valid ones by evaluating them via the imaginary reward function and the validity function $c$. All experiments are repeated with 5 seeds, and the mean and standard deviation are plotted. Further details regarding the experimental setting are reported in Apx. F.

**Bayesian reward regret.** We compare the performances of DIFFPS and DIFFPS-$N$ in terms of reward regret (see Fig. 2a) against two posterior sampling (PS) baselines. The first baseline ($m$-bandit) uses PS to solve exploration-exploitation over the given $m$-dimensional action set $\Omega$. Meanwhile, the second baseline ($D$-bandit) employs PS with the action set defined as the unit sphere in $\mathbb{R}^D$. Interestingly, as can be seen in Fig. 2a, the reward regret incurred by DIFFPS almost matches that of the bandit scheme given the true $m$-dimensional action set, and subsequently incurs low constant regret due to the approximately learned action set, as indicated by Theorem 5.1. Meanwhile, $D$-bandit incurs in a significantly higher regret due to the high dimensionality of the action space.

**Efficient reward learning.** We analyse the ability of DIFFPS and DIFFPS-$N$ to efficiently perform reward learning (see Fig. 2b) against the same baselines used to evaluated the reward regret, namely

$m$-bandit, which solves exploration-exploitation over the given $m$-dimensional action set $\Omega$, and $D$-bandit, that considers the unit sphere in $\mathbb{R}^D$ as action set. Fig. 2b shows the convergence of the reward posterior mean $\mu_t$ (of $q_t$) to the true reward model parameter $\theta_*$ for all $t \in [T]$, with respect to the distance $d(\mu_t, \theta_*) \coloneqq \|\Pi_V \mu_t - \Pi_V \theta_*\|_2 / \|\Pi_V \theta_*\|_2$ over the iterations. Once again, one can notice that DIFFPS behaves with a similar rate as $m$-bandit, although neither the low-dimensional action space $\Omega$ nor $m$ are given. This shows that DIFFPS can leverage unlabeled offline data to efficiently learn the lower dimensional reward parameter.

**Misgeneration regret and its controllability.** In Fig. 2c, we show the misgeneration regret as in Def. 2 incurred by DIFFPS and DIFFPS-$N$ given the same environment and setup as in the previous experiments. Fixed $\epsilon_c = 0.15$, the dashed black line represents the misgeneration regret obtained by a policy sampling actions $x_1, \ldots x_T$ with $c(x_t) = \epsilon_c$ for all $t$. As shown in the plot, DIFFPS achieves an average misgeneration regret smaller than $\epsilon_c = 0.15$ per iteration. Moreover, when black-box access to the validity function $c$ is available, it is possible to generate $N$ samples (here $N = 30$) at each iteration, and select the most promising valid samples. This can be done by querying $c(x)$ and selecting a sample satisfying $c(x) \leq \epsilon_c$ while achieving a reward close to $\tilde{y}_t$ w.r.t. the reward function $\tilde{r}_t$. Crucially, this procedure does not lead to higher statistical cost as the imaginary reward $\tilde{r}_t$ is known. By leveraging this, DIFFPS-$N$ achieves lower misgeneration as well as reward regret.

## 7 RELATED WORK

We review relevant work in high-dimensional bandit optimization, model-based optimization via conditional sampling, diffusion models for function optimization, and diffusion models theory.

**High-dimensional bandit and Bayesian optimization.** Many real-world black-box function optimization problems are modeled as high-dimensional bandit, including Bayesian optimization (Frazier, 2018). Typically, the high-dimensionality is addressed by either leveraging known or learned structure of the reward function (cf. Kveton et al., 2017; Lale et al., 2019; Kassraie et al., 2022), or by exploiting a known or learned representation of the action set (cf. Mutny & Krause, 2018; Griffiths & Hernández-Lobato, 2020; Wang et al., 2016; Kirschner et al., 2019; Djolonga et al., 2013), which includes VAE-based Bayesian optimization (Gómez-Bombarelli et al., 2018; Griffiths & Hernández-Lobato, 2020; Grosnit et al., 2021; Goodfellow et al., 2020). In contrast, DIFFPS directly performs black-box function optimization on the approximate data manifold using a learned diffusion sampler, without relying on a predefined or learned action space representation.

**Model-based optimization via conditional sampling and inverse modeling.** Various methods optimize a black-box function $f$ using datasets as $\{(x^i, y^i = f(x^i)\}$ through conditional sampling or inverse models. These approaches can be categorized into offline, e.g., (Uehara et al., 2024b), which use only pre-existing labeled data, and active, which can query an online oracle (e.g., Brookes et al., 2019; Kumar & Levine, 2020). Arguably, the closest work to ours is (Kumar & Levine, 2020), where the authors propose a randomized labeling strategy to approximate a posterior sampling using GANs (Goodfellow et al., 2020) and VAEs (Kingma, 2013).

**Diffusion models guidance, black-box optimization, and fine-tuning.** To steer diffusion-based generation towards designs meeting specific conditions, guidance techniques are commonly employed (Song et al., 2020; Ho & Salimans, 2022). While these methods can enhance conditional generation in DIFFPS, they are orthogonal to our work, which focuses on provably optimizing an unknown function rather than sampling predefined target values. Interestingly, our approach can be interpreted as a way to automate this process by algorithmically exploring function values to identify maxima. Additionally, some studies have used diffusion models for offline (Krishnamoorthy et al., 2023; Kong et al., 2024) and online black-box optimization (Uehara et al., 2024a; Wu et al., 2024). Unlike these approaches, which rely on upper confidence bounds (Lattimore & Szepesvári, 2020), we extend posterior sampling with diffusion models and provide both experimental (see Sec. 6) and theoretical (see Theorem 5.1) evidence that our method's statistical complexity adapts to the data intrinsic dimensionality. Moreover, unlike prior works that require a pre-trained diffusion model or labeled data, we address the case where only unlabeled offline data is available.

**Diffusion models theory.** Recent research on diffusion models theory relevant to our work falls into two categories. First, studies have established convergence rates based on the intrinsic data dimensionality under exact score estimation (e.g., De Bortoli, 2022). Second, recent works have

provided statistical guarantees for unconditional and conditional generation by accounting for score estimation and linking it to offline bandits (Chen et al., 2023; Li et al., 2024; Oko et al., 2023; Metevier et al., 2019). Building on these results, we establish guarantees for online decision-making, where an agent generates actions to navigate the exploration-exploitation trade-off with respect to an unknown reward function, leveraging offline unlabeled data to implicitly learn an action space corresponding to the data manifold.

## 8 CONCLUSIONS

In this work, we introduced a posterior sampling scheme with statistical guarantees that uses diffusion models to solve bandit optimization directly on the learned data manifold. Before concluding, we highlight a few key discussion points.

**Data-dependent guarantees for decision-making.** Theorem 5.1 states that the regret incurred by DIFFPS adapts to the intrinsic data dimensionality $m$. We believe this measure can help in bridging the gap between statistical complexity in decision-making and real-world applications, where data like molecules and proteins have intrinsic dimensions that can be estimated using known methods (Stanczuk et al., 2024; Kamkari et al., 2024; Campadelli et al., 2015; Verveer & Duin, 1995).

**Beyond bandits and diffusion** DIFFPS can be generalized beyond diffusion (see GENPS in Appendix B), and a significant part of the analysis does not rely on a specific generative model. Moreover, the algorithm and its analysis can be extended to other decision-making settings including contextual bandits (Chu et al., 2011) and reinforcement learning (Sutton et al., 1998), leading to decision-making algorithms based on generative models while preserving insightful theoretical guarantees.

To summarize, we introduced *generative bandit*, a generalization of classic bandit optimization where the action space is the unknown support of a complex data distribution, also known as *data manifold*. Furthermore, we proposed **D**iffusion **P**osterior **S**ampling (DIFFPS), an algorithm that solves the exploration-exploitation problem directly on the learned data manifold. Next, we presented regret guarantees showing how the statistical complexity of this process adapts to the *intrinsic data dimensionality* and how it depends on the available offline data. Ultimately, we have performed an experimental evaluation of the proposed algorithm supporting our theoretical claims.

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

APPENDIX

# A    LIST OF SYMBOLS

### Basic mathematical objects

| | | |
|---|---|---|
| $X^\dagger$ | $\triangleq$ | Moore-Penrose pseudo-inverse of matrix X |
| $[N]$ | $\triangleq$ | Set of integers $\{1, \ldots, N\}$ |
| $\text{supp}(P)$ | $\triangleq$ | Support of $P$, i.e., $\text{supp}(P) := \{x \in \mathbb{R}^D : P(x) > 0\}$ |
| $\|A\|_F$ | $\triangleq$ | Frobenius norm of matrix $A$ |

### (Generative) Bandit Optimization

| | | |
|---|---|---|
| $T$ | $\triangleq$ | Number of rounds or interactions |
| $t$ | $\triangleq$ | Round or interaction index, namely $t \in [T]$ |
| $\Omega$ | $\triangleq$ | Action set, if $\Omega := \text{supp}(P_x)$ then $\Omega$ corresponds with the data manifold |
| $\theta_*$ | $\triangleq$ | True reward parameter |
| $\Theta$ | $\triangleq$ | Set of reward parameters |
| $q$ | $\triangleq$ | Prior distribution on reward parameters $\Theta$, $q = q_1$ |
| $r_{\theta_*}$ | $\triangleq$ | True reward model parametrized by $\theta_*$ |
| $\pi_t$ | $\triangleq$ | (Exact) policy at time $t$, $\pi_t \in \mathcal{P}(\mathbb{R}^D)$ |
| $\pi = \{\pi_t\}_{t \in [T]}$ | $\triangleq$ | (Exact) policy |
| $x_t$ | $\triangleq$ | Action played at iteration $t \in [T]$ |
| $y_t$ | $\triangleq$ | Noisy reward observation observed at time $t$ |
| $\epsilon_t$ | $\triangleq$ | Zero-mean noise observed at time step $t \in [T]$ |
| $\nu_\theta$ | $\triangleq$ | Bandit instance with true reward parameter $\theta$ |
| $c$ | $\triangleq$ | Validity function, $c : \mathbb{R}^D \to \mathbb{R}$ |
| $\mathcal{D}_{\text{unlabeled}}$ | $\triangleq$ | Unlabeled dataset of $n$ data points, i.e., $\mathcal{D}_{\text{unlabeled}} = \{(x_i)\}_{i=1}^n$ |
| $P_x$ | $\triangleq$ | Data distribution |
| $n$ | $\triangleq$ | Number of available offline unlabeled data points, i.e., $n := |\mathcal{D}_{\text{unlabeled}}|$ |

### Generative Models and Diffusion

| | | |
|---|---|---|
| $K$ | $\triangleq$ | Terminal time of diffusion sampling process |
| $P^0(x \mid y)$ | $\triangleq$ | Initial conditional sampling distribution given $y$, i.e., $x(0) \sim P^0(x \mid y)$ |
| $P^k(x \mid y)$ | $\triangleq$ | Conditional sampling distribution at time $k$ given $y$, i.e., $x(k) \sim P^k(x \mid y)$ |
| $\nabla_x \log p^k(x \mid y)$ | $\triangleq$ | Conditional score at time $k$ |
| $\mathcal{S}$ | $\triangleq$ | Arbitrary function class to approximate score function, defined in App. E for Thr. 5.1. |
| $s$ | $\triangleq$ | Function in $\mathcal{S}$ exactly minimizing Eq. 2, i.e., exact score given realizability in Assumption 5.3 |
| $\hat{s}$ | $\triangleq$ | Approximate score function computed via approximate score matching |
| $w$ | $\triangleq$ | Wiener process |
| $\phi^k(x' \mid x)$ | $\triangleq$ | Conditional distribution of $x(k)$ given $x(0)$, i.e., $\phi^k(x' \mid x) = \mathcal{N}(\alpha(k)x, h(k)I_D)$ |
| $k_0$ | $\triangleq$ | Early-stopping time of diffusion process |
| $\ell$ | $\triangleq$ | Score matching loss function |

### Diffusion Posterior Sampling (DIFFPS)

| | | |
|---|---|---|
| $P$ | $\triangleq$ | Exact unconditional generative model distribution, i.e., $P = P_x$ and $\Omega = \text{supp}(P)$ |
| $\widehat{P}$ | $\triangleq$ | Approximate unconditional generative model distribution |
| $\widehat{\Omega}$ | $\triangleq$ | Support of approximate unconditional generative model, i.e., $\widehat{\Omega} := \text{supp}(\widehat{P})$ |
| $\tilde{\theta}_t$ | $\triangleq$ | Reward parameter sampled at iteration $t \in [T]$ of DIFFPS |
| $\tilde{r}_t$ | $\triangleq$ | Reward function sampled at iteration $t \in [T]$ of DIFFPS, i.e., $\tilde{r}_t := \tilde{r}_{\tilde{\theta}_t}$ |
| $\tilde{y}_t$ | $\triangleq$ | Maximum of imaginary reward $\tilde{r}_t$ over $\Omega$, see line 6 Alg. 1 |
| $P(\cdot \mid \tilde{y}_t)$ | $\triangleq$ | Exact conditional diffusion model given reward $\tilde{y}_t$ and reward $\tilde{r}_t$ |
| $\pi_t$ | $\triangleq$ | Exact policy at time $t$, i.e., $\pi_t := P(\cdot \mid \tilde{y}_t)$ |
| $\Omega_{\tilde{r}_t}$ | $\triangleq$ | Support of exact policy $\pi_t$, i.e., $\Omega_{\tilde{r}_t} := \text{supp}(\pi_t)$ |
| $\widehat{P}(\cdot \mid \tilde{y}_t)$ | $\triangleq$ | Approximate conditional diffusion model given reward $\tilde{y}_t$ and reward $\tilde{r}_t$ |
| $\hat{\pi}_t$ | $\triangleq$ | Approximate (sampling policy at time $t$, i.e., $\pi_t := \widehat{P}(\cdot \mid \tilde{y}_t)$ |
| $\widehat{\Omega}_{\tilde{r}_t}$ | $\triangleq$ | Support of approximate policy $\hat{\pi}_t$, i.e., $\widehat{\Omega}_{\tilde{r}_t} := \text{supp}(\hat{pi}_t)$ |

| | | |
|---|---|---|
| $P_{x,y}$ | $\triangleq$ | Joint distribution of data points $(x, y) \in \mathcal{D}$, see line 4 Alg. 1 |
| $P_{x\|y=\tilde{y}_t}$ | $\triangleq$ | Conditional distribution of $x$ given $y = \tilde{y}_t$ from $P_{x,y}$ of $(x, y) \in \mathcal{D}$, see line 4 Alg. 1 |
| $\xi_i$ | $\triangleq$ | Sample from Gaussian noise used to label $\mathcal{D}_{\text{unlabeled}}$, see line 4 Alg. 1 |
| $\nu^2$ | $\triangleq$ | Variance of noise Gaussian distribution, i.e., $\xi_i \sim \mathcal{N}(0, \nu^2)$, see line 4 Alg. 1 |
| $\mathcal{D}$ | $\triangleq$ | Dataset obtained via labeling $\mathcal{D}_{\text{unlabeled}}$, see line 4 Alg. 1 |
| $\hat{s}_t$ | $\triangleq$ | Approximate score function estimator at iteration $t \in [T]$ |

### Regret Analysis

| | | |
|---|---|---|
| $\mathcal{BR}_r(T, \pi)$ | $\triangleq$ | Bayesian reward regret, as in Definition 2 |
| $\mathcal{BR}_c(T, \pi)$ | $\triangleq$ | Bayesian misgeneration regret, as in Definition 2 |
| $D$ | $\triangleq$ | Ambient space dimensionality |
| $m$ | $\triangleq$ | Intrinsic data dimensionality, as in Definition 3 |
| $\mathcal{BR}_r^{\Omega}(T, \hat{\pi})$ | $\triangleq$ | In-manifold reward sub-optimality occurred by exact policy $\pi$, as in Prop. 1 |
| $\Delta_{(\Omega, \widehat{\Omega})}(T, \hat{\pi})$ | $\triangleq$ | In-manifold reward sub-optimality due to approximate policy, as in Prop. 1 |
| $z$ | $\triangleq$ | Latent variable, i.e., $x = Vz$ with $z \in \mathbb{R}^m$ |
| $V$ | $\triangleq$ | Matrix $V \in \mathbb{R}^{D \times m}$ such that $x = Vz$, $x \in \mathbb{R}^D$, $z \in \mathbb{R}^m$ |
| $\widehat{V}$ | $\triangleq$ | Learned approximation of matrix $V$ |
| $\Pi_V$ | $\triangleq$ | Projection onto $\Omega$, i.e., $\Pi_V := VV^T$ |
| $L$ | $\triangleq$ | Upper bound on $\|x_t\|_2$, as in Assumption 5.2 |
| $\Sigma$ | $\triangleq$ | Variance of latent distribution $P_z$ of $z$ as in Assumption 5.3 |
| $\lambda_{\min}$ | $\triangleq$ | Lower bound on eigenvalues of $\Sigma$, as in Assumption 5.3 |
| $\lambda_{\max}$ | $\triangleq$ | Upper bound on eigenvalues of $\Sigma$, as in Assumption 5.3 |
| OnlineDS | $\triangleq$ | Online distribution shift, as in Eq. 5 |
| $\bar{y}$ | $\triangleq$ | Maximum value of $\tilde{y}_t$ for $t \in [T]$, i.e., $\bar{y} := \max_{t \in [T]} \tilde{y}_t$ |
| $H_t$ | $\triangleq$ | History observed until time $t \in [T]$, i.e., $H_t := \{x_1, y_1, \ldots, x_t, y_t\}$ |
| $U_t$ | $\triangleq$ | Upper confidence bound at time $t$ |
| $L_t$ | $\triangleq$ | Lower confidence bound at time $t$ |
| $A_t$ | $\triangleq$ | $A_t := \Pi_V(\Sigma_t + \lambda I_D)\Pi_V$ for $\lambda > 0$ |
| $B_t$ | $\triangleq$ | $B_t \in \mathbb{R}^{m \times m}$ full-rank symmetric matrix s.t. $A_t = VB_tV^\top$ |
| $\sqrt{\beta_{t,\delta}}$ | $\triangleq$ | $(1 - \delta)$-probability confidence interval at time $t \in [T]$, as in Lemma D.3 |
| $DS$ | $\triangleq$ | Distribution shift, as in Eq. 19 |
| $\angle(\widehat{V}, V)$ | $\triangleq$ | Subspace angle between $V$ and $\widehat{V}$, i.e., $\angle(\widehat{V}, V) := \|\widehat{V}\widehat{V}^\top - VV^\top\|_F^2$ |
| $\tilde{\beta}_t$ | $\triangleq$ | low-dimensional parameter of $\tilde{r}_t$, $\tilde{\beta}_t \in \mathbb{R}^m$ |
| $\Psi$ | $\triangleq$ | Arbitrary function class $\Psi : \mathbb{R}^{m+1} \times [k_0, T] \to \mathbb{R}^m$ |

# B  GENERATIVE POSTERIOR SAMPLING

In this section, we first present **G**enerative **P**osterior **S**ampling (GENPS), a generative model independent meta-algorithm that generalizes Diffusion Posterior Sampling beyond diffusion models, and tackles the generative bandit problem introduced in Definition 1.

## B.1  ALGORITHM: GENERATIVE POSTERIOR SAMPLING (GENPS)

---

**Algorithm 2** GENPS: Generative Posterior Sampling (with offline unlabeled data)

---

1: **Input:** $T$ : number of online samples, $q_1$ : reward parameter prior, $\mathcal{D}_{\text{unlabeled}} = \{(x_i)\}_{i=1}^n$ : unlabeled data, $\pi$ : generative model
2: **for** $t = 1, 2, \ldots, T$ **do**
3:     Sample reward parameter $\theta_t \sim q_t$ and define $\tilde{r}_t := r_{\theta_t}$
4:     Label data in $\mathcal{D}_{\text{unlabeled}}$ via $\tilde{r}_t$: $\mathcal{D} := \{(x_i, y_i := \tilde{r}_t(x_i) + \xi_i\}_{i=1}^n$ with $\xi_i \sim \mathcal{N}(0, \nu^2)$
5:     Train conditional generative model $\hat{\pi}_t$ on $\mathcal{D}$
6:     Compute maximum imaginary reward $\tilde{y}_t = \max_{x \in \Omega} r_{\theta_t}(x)$
7:     Sample $x_t \sim \hat{\pi}_t(\cdot \mid \tilde{y}_t)$ via conditional generation
8:     Play $x_t$ and observe $y_t \sim r_{\theta_*}(x_t) + \epsilon_t$
9:     Compute $q_{t+1}$ via posterior update
10: **end for**

---

In the following, we present a detailed explanation of Algorithm 2. First, the algorithm samples an imaginary reward parameter from the rewards prior, namely $\theta_t \sim q_t$ (line 3). Then, it computes the labeled dataset $\mathcal{D}$ by labeling the dataset $\mathcal{D}_{\text{unlabeled}}$ by defining pairs $(x_i, y_i)$ with $y_i := \tilde{r}_t(x_i)$, where we define $\tilde{r}_t := r_{\theta_t}$ (line 4). Afterwards, GENPS trains a conditional generative model $\tilde{\pi}(\cdot \mid y)$ on the labeled dataset $\mathcal{D}$ (line 5), and computes the maximum imaginary reward value over $\Omega$, namely $\tilde{y}_t$ (line 6). The same observations regarding this oracle step made in Section 4 w.r.t. DIFFPS extend to GENPS. Once $\tilde{y}_t$ is computed, the algorithm approximately samples from the region of the manifold $\Omega$ achieving reward $\tilde{y}_t$, namely $\Omega_{\tilde{r}_t}$, via conditional generation $x_t \sim \hat{\pi}_t(\cdot \mid \tilde{y}_t)$ (line 7). Ultimately, it plays action $x_t$ to observe feedback $r_{\theta^*}(x_t) + \epsilon_t$ (line 8), and performs posterior update of the reward prior $q_t$ (line 9) to integrate the new evidence gained about the true reward function $r_{\theta_*}$.

## B.2  EXTENSION OF RESULTS OF DIFFPS TO GENPS

Interestingly, the argument for approximate in-manifold exploration shown in Equation 4 w.r.t. DIFFPS extends to GENPS, and analogously also the regret decomposition Proposition 1. Nonetheless, while the $\mathcal{BR}_r^\Omega(T, \hat{\pi})$ of the reward regret can be bounded analogously for GENPS, the term $\Delta_{(\Omega, \widehat{\Omega})}(T, \hat{\pi})$, as well as the validity regret, require generative model specific estimation guarantees and therefore the regret results presented in Theorem 5.1 does not trivially extend to Algorithm 2.

## C  POSTERIOR UPDATES

**Posterior Sampling.** Given reward prior $q_t = \mathcal{N}(\mu_t, \Sigma_t)$, we compute the posterior $q_{t+1}$ using the standard closed-form updates for Gaussians given by (Russo et al., 2020):

$$\Sigma_{t+1} = \left(\Sigma_t + x_t x_t^\top / \sigma^2\right)^{-1} \quad \text{and} \quad \mu_{t+1} = \Sigma_{t+1} \left(\Sigma_t^{-1}\mu_t + x_t(y_t + \epsilon_t)/\sigma^2\right)^{-1} \tag{6}$$

where $(\mu_t, \Sigma_t)$ are the prior mean and covariance, respectively, and $\epsilon_t \sim \mathcal{N}(0, \sigma^2)$.

## D   GENERATIVE (BAYESIAN) REGRET ANALYSIS

First, we state the following decomposition result for the (Bayesian) reward regret as presented in Definition 2.

### D.1   (BAYESIAN) REWARD REGRET DECOMPOSITION

**Proposition 1** (Bayesian reward regret decomposition). *Given a policy $\hat{\pi}$ corresponding to running Algorithm 2, we have:*

$$
\mathcal{BR}_r(T, \hat{\pi}) \leq \underbrace{\sum_{t=1}^{T} \mathop{\mathbb{E}}_{\theta_* \sim q} \mathop{\mathbb{E}}_{x_t \sim \pi_t} |r_*(x^*) - r_*(x_t)|}_{\mathcal{BR}_r^\Omega(T, \hat{\pi})} + \underbrace{\sum_{t=1}^{T} \mathop{\mathbb{E}}_{\theta_* \sim q} \left| \mathop{\mathbb{E}}_{x_t \sim \hat{\pi}_t} [r_*(x_t)] - \mathop{\mathbb{E}}_{x_t \sim \pi_t} [r_*(x_t)] \right|}_{\Delta_{(\Omega, \hat{\Omega})}(T, \hat{\pi})}
$$

*Proof.* First, recall the definition of (Bayesian) reward regret associated to a policy $\hat{\pi}$ interacting for $T$ steps with a problem instance $\theta^* \sim q$, namely:

$$
\mathcal{BR}_r(T, \hat{\pi}) \coloneqq \mathop{\mathbb{E}}_{\theta_* \sim q} \left[ \sum_{t=1}^{T} r_{\theta_*}(x^*) - \mathop{\mathbb{E}}_{x_t \sim \hat{\pi}_t} [r_{\theta_*}(x_t)] \right]
$$

To derive the decomposition result we start by writing:

$$
\mathop{\mathbb{E}}_{x_t \sim \hat{\pi}_t} [r_{\theta_*}(x_t)] \geq \mathop{\mathbb{E}}_{x_t \sim \pi_t} [r_{\theta_*}(x_t)] - \left| \mathop{\mathbb{E}}_{x_t \sim \hat{\pi}_t} [r_{\theta_*}(x_t)] - \mathop{\mathbb{E}}_{x_t \sim \pi_t} [r_{\theta_*}(x_t)] \right|
$$

$$
= r_{\theta_*}(x^*) - \mathop{\mathbb{E}}_{x_t \sim \pi_t} \left| r_{\theta_*}(x^*) - r_{\theta_*}(x_t) \right| - \left| \mathop{\mathbb{E}}_{x_t \sim \hat{\pi}_t} [r_{\theta_*}(x_t)] - \mathop{\mathbb{E}}_{x_t \sim \pi_t} [r_{\theta_*}(x_t)] \right|
$$

then by defining $l_t$ s.t. $\mathcal{BR}_r(T, \hat{\pi}) = \mathbb{E}_{\theta^* \sim q} \left[ \sum_{t=1}^{T} l_{t, \theta^*} \right]$, we have:

$$
l_{t, \theta^*} \leq \mathop{\mathbb{E}}_{x_t \sim \pi_t} \left| r_{\theta_*}(x^*) - r_{\theta_*}(x_t) \right| + \left| \mathop{\mathbb{E}}_{x_t \sim \hat{\pi}_t} [r_{\theta_*}(x_t)] - \mathop{\mathbb{E}}_{x_t \sim \pi_t} [r_{\theta_*}(x_t)] \right|, \tag{7}
$$

which leads to:

$$
\mathcal{BR}_r(T, \hat{\pi}) = \mathop{\mathbb{E}}_{\theta^* \sim q} \left[ \sum_{t=1}^{T} l_{t, \theta^*} \right]
$$

$$
\leq \mathop{\mathbb{E}}_{\theta_* \sim q} \left[ \sum_{t=1}^{T} \mathop{\mathbb{E}}_{x_t \sim \pi_t} \left| r_{\theta_*}(x^*) - r_{\theta_*}(x_t) \right| + \left| \mathop{\mathbb{E}}_{x_t \sim \hat{\pi}_t} [r_{\theta_*}(x_t)] - \mathop{\mathbb{E}}_{x_t \sim \pi_t} [r_{\theta_*}(x_t)] \right| \right]
$$

$$
\leq \sum_{t=1}^{T} \mathop{\mathbb{E}}_{\theta_* \sim q} \mathop{\mathbb{E}}_{x_t \sim \pi_t} \left| r_{\theta_*}(x^*) - r_{\theta_*}(x_t) \right| + \sum_{t=1}^{T} \mathop{\mathbb{E}}_{\theta_* \sim q} \left| \mathop{\mathbb{E}}_{x \sim \hat{\pi}_t} [r_{\theta_*}(x_t)] - \mathop{\mathbb{E}}_{x_t \sim \pi_t} [r_{\theta_*}(x_t)] \right|
$$

$\square$

### D.2   BOUNDING THE (BAYESIAN) REWARD REGRET $\mathcal{BR}_r(T, \hat{\pi})$

Given the decomposition result in Proposition 1 for the reward regret, in the following we proceed by upper bounding separately the terms $\mathcal{BR}_r^\Omega(T, \hat{\pi})$ and $\Delta_{(\Omega, \hat{\Omega})}(T, \hat{\pi})$.

#### D.2.1   UPPER BOUND $\mathcal{BR}_r^\Omega(T, \hat{\pi})$

We now proceed upper bounding the term $\mathcal{BR}_r^\Omega(T, \hat{\pi})$, which captures the regret incurred by the agent by generating samples within the true manifold $\Omega$ with the exact policy $\pi$. In fact, notice that $\mathcal{BR}_r^\Omega(T, \hat{\pi})$ does not depend on the approximate policy $\hat{\pi}$, but only on the exact policy $\pi$. First, we state the following decomposition result which extends (Russo & Van Roy, 2014, Proposition 1) to the case of generative, hence stochastic and approximate, policies.

**Proposition 2** (Decomposition PS regret on manifold). *Given a policy $\hat{\pi}$ corresponding to running Algorithm 1, for any upper confidence sequence $\{U_t \mid t \in \mathbb{N}\}$ defined as in (Russo & Van Roy, 2014, Section 4.1), we have that:*

$$\mathcal{BR}_r^{\Omega}(T, \hat{\pi}) = \sum_{t=1}^{T} \underset{\theta_* \sim q}{\mathbb{E}} \underset{x_t \sim \pi_t}{\mathbb{E}} [U_t(x_t) - r_{\theta_*}(x_t)] + \sum_{t=1}^{T} \underset{\theta_* \sim q}{\mathbb{E}} [r_{\theta_*}(x^*) - U_t(x^*)]$$

*Proof.* For each term $t \in [T]$ within the sum in $\mathcal{BR}_r^{\Omega}(T, \hat{\pi})$ defined as in Proposition 1, we have:

$$\underset{\theta_* \sim q}{\mathbb{E}} \underset{x_t \sim \pi_t}{\mathbb{E}} [r_{\theta_*}(x^*) - r_{\theta_*}(x_t)] \overset{(1)}{=} \underset{\theta_* \sim q}{\mathbb{E}} \underset{x_t \sim \pi_t}{\mathbb{E}} [r_{\theta_*}(x^*) - r_{\theta_*}(x_t) \mid H_t]$$

$$= \underset{\theta_* \sim q}{\mathbb{E}} \underset{x_t \sim \pi_t}{\mathbb{E}} [U_t(x_t) - U_t(x_t) + r_{\theta_*}(x^*) - r_{\theta_*}(x_t) \mid H_t]$$

$$\overset{(2)}{=} \underset{\theta_* \sim q}{\mathbb{E}} \underset{x_t \sim \pi_t}{\mathbb{E}} [U_t(x_t) - U_t(x^*) + r_{\theta_*}(x^*) - r_{\theta_*}(x_t) \mid H_t]$$

$$= \underset{\theta_* \sim q}{\mathbb{E}} \underset{x_t \sim \pi_t}{\mathbb{E}} [U_t(x_t) - r_{\theta_*}(x_t) \mid H_t] + \underset{\theta_* \sim q}{\mathbb{E}} [r_{\theta_*}(x^*) - U_t(x^*) \mid H_t]$$

$$\overset{(3)}{=} \underset{\theta_* \sim q}{\mathbb{E}} \underset{x_t \sim \pi_t}{\mathbb{E}} [U_t(x_t) - r_{\theta_*}(x_t)] + \underset{\theta_* \sim q}{\mathbb{E}} [r_{\theta_*}(x^*) - U_t(x^*)]$$

Where in step (1) we use the law of total expectation with history $H_t \coloneqq \{x_1, y_1, \ldots, x_t, y_t\}$, in step (2) we employ Lemma D.1, and in step (3) we use again the law of total expectation in the reverse direction. Ultimately, summing over $t \in [T]$ leads to the result in the statement. $\square$

In classic posterior sampling (Russo & Van Roy, 2014), given $\theta_t \sim q_t$, the action selected is deterministically chosen as $x_t \in \arg\max_{x \in \mathcal{X}} r_{\theta_t}(x)$. On the other hand, DIFFPS first computes deterministically $\tilde{y}_t \in \max_{x \in \Omega} r_{\theta_t}(x)$ and then approximately samples $x_t \sim \hat{\pi} = \widehat{P}(\cdot \mid \tilde{y}_t)$ via a generative (diffusion) process. Nonetheless, notice that due to the decomposition result in Proposition 1, the random variable $x_t$ within the definition of $\mathcal{BR}_r^{\Omega}(T, \hat{\pi})$ is an imaginary random variable introduced for the sake of analysis and sampled according to the exact policy $\pi_t = P(\cdot \mid \tilde{y}_t)$. This is a crucial observation to prove the following Lemma used within the proof of Proposition 2 in step (2).

**Lemma D.1** (Generative action replacement). *Given the notation above, we can state the following:*

$$\underset{\theta_* \sim q}{\mathbb{E}} \underset{x_t \sim \pi_t}{\mathbb{E}} [U_t(x_t) \mid H_t] = \underset{\theta_* \sim q}{\mathbb{E}} \underset{x_t \sim \pi_t}{\mathbb{E}} [U_t(x^*) \mid H_t] \tag{8}$$

*Proof.* Recall that $x_t \sim \pi_t = P(\cdot \mid \tilde{y}_t = \max_{x \in \Omega} r_{\theta_t}(x))$. Since $P$ is the exact distribution rather than the approximate distribution $\widehat{P}$, we have that $x \in \arg\max_{x \in \Omega} r_{\theta_t}(x)$ with $\theta_t \sim q_t$. Meanwhile, notice that we can characterize $x^*$ as $x^* \sim \pi^* = P^*(\cdot \mid y^* = \max_{x \in \Omega} r_{\theta_*}(x))$ and therefore $x^* \in \arg\max_{x \in \Omega} r_{\theta_*}(x)$ with $\theta_* \sim q = q_0$. Hence we can see that the exact sampling process can be seen as an implementation of the argmax operation and therefore both $U_t(x_t)$ and $U_t(x^*)$ can be seen as obtained via the sampling process of $\theta_t$ and $\theta_*$ respectively, plus a deterministic operation, i.e., the argmax. As a consequence, by conditioning on $H_t$ we have that $\theta_t$ and $\theta_*$ are identically distributed and since $U_t(x_t)$ and $U_t(x^*)$ are deterministic given $\theta_t$ and $\theta_*$, then they are identically distributed as well given $H_t$ as it is the case in the classic posterior sampling analysis, e.g., (Russo & Van Roy, 2014, Section 5.2, Proposition 1). $\square$

We now upper bound the term $\mathcal{BR}_r^{\Omega}(T, \hat{\pi})$ via an optimistic analysis leveraging Assumption 5.1 stating that $\Omega$ is a low-dimensional linear subspace, and Assumption 5.2 stating the fact that the reward is representable via a linear model.

**Lemma D.2** (Upper bound $\mathcal{BR}_r^{\Omega}(T, \hat{\pi})$: in-manifold regret given exact generative model). *Given a policy $\hat{\pi}$ corresponding to running Algorithm 1, and Assumptions 5.2, 5.1 we have:*

$$\mathcal{BR}_r^{\Omega}(T, \hat{\pi}) = \widetilde{O}(m\sqrt{T}) \tag{9}$$

*Proof.* First, recall the following decomposition of $\mathcal{BR}_r^\Omega(T, \hat{\pi})$ given by Proposition 2.

$$\mathcal{BR}_r^\Omega(T, \hat{\pi}) = \sum_{t=1}^T \mathop{\mathbb{E}}_{\theta_* \sim q} \mathop{\mathbb{E}}_{x_t \sim \pi_t} [U_t(x_t) - r_{\theta_*}(x_t)] + \sum_{t=1}^T \mathop{\mathbb{E}}_{\theta_* \sim q} [r_{\theta_*}(x^*) - U_t(x^*)] \tag{10}$$

For $r_{\theta_*}$ taking values in $[0, R] \subseteq [-C, C]$ this implies:

$$\mathcal{BR}_r^\Omega(T, \hat{\pi}) \leq \underbrace{\sum_{t=1}^T \mathbb{E}[U_t(x_t) - L_t(x_t)]}_{\phi} + \underbrace{2R \sum_{t=1}^T [\mathbb{P}(r_{\theta_*}(x^*) > U_t(x^*)) + \mathbb{P}(r_{\theta_*}(x_t) < L_t(x_t))]}_{\psi} \tag{11}$$

where $U_t$ and $L_t$ are upper and lower confidence bounds $L_t : \mathcal{X} \to \mathbb{R}$ and $U_t : \mathcal{X} \to \mathbb{R}$ so that $L_t(x) \leq r_{\theta_*}(x) \leq U_t(x)$ w.h.p. for all $x$ and $t$. As in a typical optimistic analysis, we build a ellipsoidal confidence set $\Theta_t$ and define $U_t := \max\{R, \max_{\theta \in \Theta_t} \theta^\top x\}$ and $L_t := \min\{-R, \min_{\theta \in \Theta_t} \theta^\top x\}$. Then we will bound $\phi$ by building a valid upper bound of $\sum_{t=1}^T [U_t(x_t) - L_t(x_t)]$ for any sequence of actions, and we will bound $\psi$ by $4R$ by a proper definition of $\Theta_t$ and therefore of $U_t$ and $L_t$.

**Upper bound $\phi$** First, we introduce the following objects:

$$\Pi_V := VV^\top \qquad\qquad \text{(projection onto } \Omega)$$

$$\Sigma_t := \sum_{i=1}^t x_i x_i^\top = X_t X_t^\top$$

$$A_t := \Pi_V(\Sigma_t + \lambda I_D)\Pi_V \text{ for } \lambda > 0$$

$$B_t \in \mathbb{R}^{m \times m} \text{ full-rank symmetric matrix s.t. } A_t = V B_t V^\top$$

Then, we bound the $t$-th element within the sum in $\phi$ as follows.

$$\begin{aligned}
\phi_t &= \mathbb{E}|U_t(x_t) - L_t(x_t)| \\
&\overset{(4)}{\leq} 2\mathbb{E}|U_t(x_t) - r_{\theta_*}(x_t)| \\
&\overset{(5)}{=} 2\mathbb{E}|\tilde{\theta}_t^\top x_t - \theta_*^\top x_t| \\
&\leq \mathbb{E}\|x_t\|_{A_{t-1}^\dagger} \cdot \|\theta_* - \tilde{\theta}_t\|_{A_{t-1}} \\
&\overset{(6)}{\leq} 2\mathbb{E}\|x_t\|_{A_{t-1}^\dagger} \cdot \sqrt{\beta_{t,\delta}}
\end{aligned} \tag{12}$$

Where in step (4) we used the definition of $U_t$ and $L_t$, in step (5) we used Assumption 5.2, and in step (6) we employed Lemmata D.4 and D.3. We proceed bounding the first term within Equation 12. We have:

$$\mathbb{E}\|x_t\|_{A_{t-1}^\dagger} \overset{(7)}{\leq} \min\{1, \mathbb{E}\|x_t\|_{A_{t-1}^\dagger}\}$$

$$\overset{(8)}{=} \min\{1, \mathbb{E}\|V^\top x_t\|_{B_{t-1}^{-1}}\}$$

where in step (7) we use the fact that $l_t \leq 1$, and in step (8) we have used the definition of $A_t$ and $B_t$. Now we can bound the sum of such contributions as:

$$\sum_{t=1}^T \min\{1, \mathbb{E}\|V^\top x_t\|_{B_{t-1}^{-1}}\} \leq 2m \log\left(1 + \frac{TL^2}{m\lambda}\right)$$

by using Lemma D.5. We can now bound $\phi$ as:

$$\phi = \sum_{t=1}^{T} \phi_t$$

$$\overset{(9)}{\leq} \sqrt{T \sum_{t=1}^{T} \phi_t^2}$$

$$\overset{(10)}{\leq} 2\sqrt{T\beta_{T,\delta} \sum_{t=1}^{T} \min\{1, \mathbb{E}\, \|V^\top x_t\|_{B_t^{-1}}\}}$$

$$\overset{(11)}{\leq} 2\sqrt{T\beta_{T,\delta} 2m \log\left(1 + \frac{TL^2}{m\lambda}\right)}$$

where in step (9) we used Cauchy-Schwarz, in step (10) we used the fact that $\beta_{T,\delta} \geq \beta_{t,\delta} \forall t \in [T]$ and in step (11) we leveraged Lemma D.5. Here $\sqrt{\beta_{T,\delta}} := R\sqrt{m \log\left(\frac{1+TL^2/\lambda}{\delta}\right)} + \sqrt{\lambda}$ as stated in Lemma D.3. By plugging $\beta_{T,\delta}$ into the expression above one obtains that with probability at least $1 - \delta$:

$$\phi \leq 2\left(R\sqrt{m \log\left(\frac{1+TL^2/\lambda}{\delta}\right)} + \sqrt{\lambda}\right) \sqrt{T 2m \log\left(1 + \frac{TL^2}{m\lambda}\right)} = \tilde{O}\left(m\sqrt{T}\right)$$

**Upper bound $\psi$** By construction of the sequence of confidence intervals $\beta_{t,\delta}$ as in Lemma D.3, we have that $\mathbb{P}(\theta \notin \Theta_t \mid H_t) \leq 1/T$ and therefore $\psi \leq 4R$ as argued in (Russo & Van Roy, 2014, Section 6.2.1). $\qquad\square$

**Lemma D.3** (Confidence Intervals for $m$-dimensional linear bandits). *Given the same assumption of Theorem 5.1, for any $\delta > 0$, with probability at least $1 - \delta$ for all $t \in [T]$ we have that $\theta_*$ lies in the set:*

$$\Theta_t = \left\{\theta \in R^m : \|\hat{\theta}_t - \theta\|_{A_t} \leq \sqrt{\beta_{t,\delta}} := R\sqrt{m \log\left(\frac{1+tL^2/\lambda}{\delta}\right)} + \sqrt{\lambda}\right\} \qquad (13)$$

*Proof.* This result can be proved analogously to (Lale et al., 2019, Theorem 3) but given knowledge of the projection operator $\Pi_V = VV^\top$, thus leading to the same result as in classic $m$-dimensional linear bandits, e.g., (Abbasi-Yadkori et al., 2011, Theorem 2). $\qquad\square$

**Lemma D.4** (Subspace Cauchy–Schwarz).
$$|\tilde{\theta}_t^\top x_t - \theta_*^\top x_t| \leq \|x_t\|_{A_t^\dagger} \cdot \|\theta_* - \tilde{\theta}_t\|_{A_t} \qquad (14)$$

*Proof.* We can write:

$$|\tilde{\theta}_t^\top x_t - \theta_*^\top x_t| \overset{(12)}{=} |\tilde{\theta}_t^\top (\Pi_V x_t) - \theta_*^\top (\Pi_V x_t)|$$

$$= |(\Pi_V x_t)^\top (\tilde{\theta}_t - \theta_*)|$$

$$= |(\Pi_V x_t)^\top (A_t^\dagger)^{\frac{1}{2}} A_t^{\frac{1}{2}} (\tilde{\theta}_t - \theta_*)|$$

$$= |[(A_t^\dagger)^{\frac{1}{2}} \Pi_V x_t]^\top A_t^{\frac{1}{2}} (\tilde{\theta}_t - \theta_*)|$$

$$\overset{(13)}{\leq} \|(A_t^\dagger)^{\frac{1}{2}} \Pi_V x_t\| \cdot \|A_t^{\frac{1}{2}} (\tilde{\theta}_t - \theta_*)\|$$

$$\overset{(14)}{=} \|\Pi_V x_t\|_{A_t^\dagger} \cdot \|A_t^{\frac{1}{2}} (\tilde{\theta}_t - \theta_*)\|$$

$$\overset{(15)}{=} \|x_t\|_{A_t^\dagger} \cdot \|A_t^{\frac{1}{2}} (\tilde{\theta}_t - \theta_*)\|$$

$$\overset{(16)}{=} \|x_t\|_{A_t^\dagger} \cdot \|\theta_* - \tilde{\theta}_t\|_{A_t}$$

where step (12) in due to $x_t \sim \pi_t$ and $\mathrm{supp}(\pi_t) \subseteq \Omega$, in step (13) we used Cauchy-Schwarz, and in step (14) we have used that

$$
\begin{aligned}
\|(A_t^\dagger)^{\frac{1}{2}} \Pi_V x_t\| &= \sqrt{[(A_t^\dagger)^{\frac{1}{2}} \Pi_V x_t]^\top (A_t^\dagger)^{\frac{1}{2}} (\Pi_V x_t)} \\
&= \sqrt{(\Pi_V x_t)^T (A_t^\dagger)^{\frac{1}{2}} (A_t^\dagger)^{\frac{1}{2}} (\Pi_V x_t)} \\
&= \sqrt{(\Pi_V x_t)^\top A_t^\dagger (\Pi_V x)} \\
&= \|\Pi_V x_t\|_{A_t^\dagger},
\end{aligned}
$$

in step (15) we have used the fact that $x_t = \Pi_V x_t$ and in step (16) we have used the following:

$$
\begin{aligned}
\|A_t^{\frac{1}{2}} (\tilde{\theta}_t - \theta_*)\| &= \sqrt{[A_t^{\frac{1}{2}} (\tilde{\theta}_t - \theta_*)]^\top [A_t^{\frac{1}{2}} (\tilde{\theta}_t - \theta_*)]} \\
&= \sqrt{(\tilde{\theta}_t - \theta_*)^\top A_t^{\frac{1}{2}} A_t^{\frac{1}{2}} (\tilde{\theta}_t - \theta_*)} \\
&= \sqrt{(\tilde{\theta}_t - \theta_*)^\top A_t (\tilde{\theta}_t - \theta_*)} \\
&= \|\tilde{\theta}_t - \theta_*\|_{A_t}
\end{aligned}
$$

$\square$

**Lemma D.5** (Projected potential lemma in expectation). *Given the same assumptions of Theorem 5.1, we have:*

$$
\sum_{t=1}^{T} \min\{1, \mathbb{E} \|V^\top x_t\|_{B_{t-1}^{-1}}^2\} \le 2m \log\left(1 + \frac{TL^2}{m\lambda}\right) \tag{15}
$$

*Proof.* We first prove the result without the expectation in the LHS, for any sequence of iterates $x_t$, and then use it to upper bound the expression with expectation as in the statement. For $t \ge 1$ we have:

$$
\begin{aligned}
\det(B_t) &= \det\left(B_{t-1} + V^\top x_t x_t^\top V\right) \\
&= \det\left(B_{t-1}^{1/2}(I_m + B_{t-1}^{-1/2} V^\top x_t x_t^\top V B_{t-1}^{-1/2}) B_{t-1}^{1/2}\right) \\
&= \det(B_{t-1}) \det\left(1 + \|V^\top x_t\|_{B_{t-1}^{-1}}^2\right) \\
&= \lambda^m \prod_{i=1}^{t} \left(1 + \|V^\top x_i\|_{B_{i-1}^{-1}}^2\right)
\end{aligned}
$$

Hence for $t = T$:

$$
\sum_{i=1}^{T} \log\left(1 + \|V^\top x_i\|_{B_{i-1}^{-1}}^2\right) = \log\left(\frac{\det(B_T)}{\lambda^m}\right)
$$

$$
\le m \log\left(1 + \frac{TL^2}{m\lambda}\right)
$$

where the last step is due to (Lale et al., 2019, Lemma 11). Ultimately, we use the fact that $\min\{1, u\} \le 2\log(1 + u)$ to obtain:

$$
\sum_{t=1}^{T} \min\{1, \|V^\top x_t\|_{B_{t-1}^{-1}}^2\} \le 2m \log\left(1 + \frac{TL^2}{m\lambda}\right) \tag{16}
$$

Due to the definition of the expectation one can then upper bound the LHS in the statement with the bound in Equation (16) as it holds for any sequence of $x_t$.

$\square$

### D.2.2 Upper bound $\Delta_{(\Omega,\widehat{\Omega})}$

We now proceed upper bounding the term $\Delta_{(\Omega,\widehat{\Omega})}$, which captures the regret incurred in-manifold due to the approximate diffusion model sampling.

**Lemma D.6** (Upper bound $\Delta_{(\Omega,\widehat{\Omega})}$: in-manifold regret due to approximate generative model). *Given a policy $\hat{\pi}$ corresponding to running Algorithm 1, and given the same assumptions of Theorem 5.1, we have:*

$$\Delta_{(\Omega,\widehat{\Omega})} \leq T \cdot \mathrm{DS}(\bar{y}) \left( \frac{m^2 D + D^2 d}{n} \right)^{\frac{1}{6}} \cdot \bar{y} \tag{17}$$

*where $\bar{y} := \max_{t \in [T]} \tilde{y}_t$.*

*Proof.* Recall that:

$$\Delta_{(\Omega,\widehat{\Omega})} = \sum_{t=1}^{T} \mathbb{E}_{\theta_* \sim q} \left| \mathbb{E}_{x_t \sim \hat{\pi}_t} [r_{\theta_*}(x_t)] - \mathbb{E}_{x_t \sim \pi_t} [r_{\theta_*}(x_t)] \right|$$

From Li et al. (2024) we know that $\forall t \in [T]$, we have:

$$\mathbb{E}_{\theta_* \sim q} \left| \mathbb{E}_{x_t \sim \hat{\pi}_t} [r_{\theta_*}(x_t)] - \mathbb{E}_{x_t \sim \pi_t} [r_{\theta_*}(x_t)] \right| \leq \mathrm{DistShift}(\tilde{y}_t) \left( \frac{m^2 D + D^2 m}{n} \right)^{\frac{1}{6}} \cdot \tilde{y}_t \tag{18}$$

where $\mathrm{DS}(\tilde{y}_t)$ is defined as follows. Given the imaginary reward $\tilde{r}_t$, and labeled dataset $D_t = \{(x_i, y_i = \tilde{r}_t(x_i) + \xi_i)\}_{i \in [n]}$, we denote with $P_{x,y}$ the joint distribution such that $(x_i, y_i) \sim P_{x,y}$. And given $\tilde{y}_t$, we define the conditional distribution of $x$ given $\tilde{y}_t$ as $P_{x|y=\tilde{y}_t}$, then we have:

$$\mathrm{DS}^2(\tilde{y}_t) := \frac{\mathbb{E}_{P_{x|y=\tilde{y}_t}}[\ell(x, \tilde{y}_t; \hat{s})]}{\mathbb{E}_{P_{x,y}}[\ell(x, y; \hat{s})]} \tag{19}$$

We now define the following online distribution shift:

$$\mathrm{OnlineDS}^2(t') := \max_{t \in [t']} \mathrm{DS}^2(\tilde{y}_t) = \max_{t \in [t']} \frac{\mathbb{E}_{P_{x|y=\tilde{y}_t}}[\ell(x, \tilde{y}_t; \hat{s})]}{\mathbb{E}_{P_{x,y}}[\ell(x, y; \hat{s})]}$$

Therefore, we can upper bound the expression above as follows.

$$\Delta_{(\Omega,\widehat{\Omega})} \leq T \cdot \mathrm{OnlineDS}(T) \left( \frac{m^2 D + D^2 m}{n} \right)^{\frac{1}{6}} \cdot \bar{y}$$

where $\bar{y} := \max_{t \in [T]} \tilde{y}_t$. $\qquad\square$

### D.3 Bounding the (Bayesian) misgeneration regret

**Lemma D.7** (Bayesian misgeneration regret upper bound). *Given a policy $\hat{\pi}$ corresponding to running Algorithm 1, and given the same assumptions of Theorem 5.1, we have:*

$$\mathcal{BR}_c(T, \hat{\pi}) = \widetilde{O}\left( T \left( \sqrt{k_0 D} + \sqrt{\frac{1}{\lambda_{min}} \sqrt{\frac{Dm^2 + D^2 m}{n}} \cdot \sqrt{\frac{\bar{y}^2}{\|\beta_t\|_{\Sigma}} + m}} \right) \right)$$

*Proof.* Recall that:

$$\mathcal{BR}_c(T, \hat{\pi}) := \sum_{t=1}^{T} \mathbb{E}_{x \sim \hat{\pi}_t} [c(x)] \tag{20}$$

Given assumptions 5.1, 5.3, 5.2 and recalling that $\hat{\pi}_t := \widehat{P}(\cdot \mid \tilde{y}_t)$, we can upper bound an element of the sum within Equation 20 as in (Li et al., 2024, Theorem 6.2), obtaining:

$$\mathbb{E}_{x \sim \hat{\pi}_t} [c(x)] = O\left( \sqrt{k_0 D} + \sqrt{\angle(\widehat{V}, V)} \cdot \sqrt{\frac{\tilde{y}_t^2}{\|\tilde{\beta}_t\|_{\Sigma}} + m} \right) \tag{21}$$

where $\tilde{\beta}_t \in \mathbb{R}^m$ is the low-dimensional parameter of $\tilde{r}_t$, namely for $x \in \Omega$ we have $\tilde{r}_t(x) := \tilde{\theta}_t^\top x = \tilde{\theta}_t^\top (\Pi_V x) = (\Pi_V \theta_t)^\top x = \tilde{\beta}_t^\top z$ . Formally, by defining $\bar{y} := \max_{t \in [T]} \tilde{y}_t$, and $\bar{\beta} := \min_{t \in [T]} \|\tilde{\beta}_t\|_\Sigma$, we have

$$\sum_{t=1}^{T} \mathbb{E}_{x \sim \hat{\pi}_t} [c(x)] = O\left(T\left(\sqrt{k_0 D} + \sqrt{\angle(\widehat{V}, V)} \cdot \sqrt{\frac{\bar{y}^2}{\bar{\beta}} + m}\right)\right) \tag{22}$$

where $\angle(\widehat{V}, V)$ is the subspace angle between matrices $\widehat{V}$ and $V$. Here matrix $\widehat{V}$ represents the representation matrix implicitly learned by the diffusion model, while $V$ is the matrix representing the ground truth subspace. Formally, $\angle(\widehat{V}, V)$ measures the column space difference between $\widehat{V}$ and $V$, and is defined as:

$$\angle(\widehat{V}, V) := \|\widehat{V}\widehat{V}^\top - VV^\top\|_F^2$$

We can derive the statement by recalling that by (Li et al., 2024, Theorem 5.4), we have:

$$\angle(\widehat{V}, V) = \widetilde{O}\left(\frac{1}{\lambda_{min}}\sqrt{\frac{\mathcal{N}(\mathcal{S}, 1/n)D}{n}}\right) = \widetilde{O}\left(\frac{1}{\lambda_{min}}\sqrt{\frac{Dm^2 + D^2 m}{n}}\right) \tag{23}$$

$\square$

### D.4 (BAYESIAN) REGRET THEOREM

We can now state an upper bound on the Bayesian regret.

**Theorem 5.1** (Bayesian reward and misgeneration regret upper bound). *Given a policy $\hat{\pi}$ corresponding to running Algorithm 1 and the assumptions stated above, by choosing $k_0 = ((Dm^2 + D^2 m)/n)^{1/6}$, $\nu = 1/\sqrt{D}$, and $D \geq m^2$, defining $\bar{y} := \max_{t \in [T]} \tilde{y}_t$, we have:*

$$\mathcal{BR}_r(T, \hat{\pi}) = \widetilde{O}\left(m\sqrt{T} + T \cdot \text{OnlineDS}(T)\left(\frac{m^2 D + D^2 m}{n}\right)^{\frac{1}{6}} \cdot \bar{y}\right) \quad \text{(reward regret)}$$

$$\mathcal{BR}_c(T, \hat{\pi}) = \widetilde{O}\left(T\left(\sqrt{k_0 D} + \sqrt{\frac{mD}{n^{1/2}}} \cdot \sqrt{\bar{y}^2 + m}\right)\right) \quad \text{(misgeneration regret)}$$

*where $\text{OnlineDS}(T)$ is defined in Eq. 5.*

*Proof.* $\mathcal{BR}_r(T, \hat{\pi})$ is bounded as shown within Section D.2 and $\mathcal{BR}_c(T, \hat{\pi})$ is bounded as in Section D.3. $\square$

# E   SCORE NETWORK FUNCTION CLASS

For the sake of analysis, we consider the neural networks model class $\mathcal{S}$ with $m$-dimensional encoder-decoder structure to approximate the score function, as defined in (Li et al., 2024, Equation 4.8), namely:

$$\mathcal{S} = \left\{ s_{V,\psi}(x, y, k) = \frac{1}{h(k)} (V \cdot \psi(V^\top x, y, k) - x) : V \in \mathbb{R}^{D \times m}, \psi \in \Psi : \mathbb{R}^{m+1} \times [k_0, T] \to \mathbb{R}^m \right\}$$

where $V$ is a matrix with orthonormal columns and $\Psi$ is an arbitrary function class. Notice that a score network function class with encoder-decoder structure as $\mathcal{S}$ was first proposed by Chen et al. (2023) to derive statistical complexities for unconditional generation via diffusion models.

## F  PRACTICAL IMPLEMENTATION AND EXPERIMENTAL DETAILS

### F.1  APPROXIMATE ORACLE IMPLEMENTATIONS

In the following, we propose two practical methods to approximately implement the oracle step (line 6) in Algorithm 1.

**In-dataset maximizer.** One classic method typically used in optimization via inverse model consist in selecting the in-dataset maximizer (Krishnamoorthy et al., 2023; Kumar & Levine, 2020). Namely:

$$\tilde{y}_t = \max_{x \in \mathcal{D}} \tilde{r}_t(x)$$

In this way, $\tilde{y}_t$ can be computed efficiently, namely linearly in $n$, and by using a *best-of-N* scheme for sampling via diffusion, as discussed below, it is possible to generate actions $x_t$ better w.r.t. the imaginary reward $\tilde{r}_t$ than the ones already present in the dataset.

**Binary search on output space.** In principle, the oracle step consists in an output-maximization problem over an unknown set $\Omega$. Given enough and well distributed unlabeled data the diffusion model support $\widehat{\Omega} := \mathrm{supp}(\widehat{P})$ approximates well $\Omega$, namely $\widehat{\Omega} \approx \Omega$. Then one can perform approximate maximization over the output space of $\tilde{r}_t$ considering the domain $\Omega$ via the following scheme:

---

**Algorithm 3** Approximate binary search oracle implementation

---

1: **Input:** $\epsilon_1$ : search stopping condition, $\epsilon_2$ : validity oracle approximation, $\epsilon_3$ : sampling approximation, $R_{\max}$ : upper bound reward function, $\tilde{r}_t$ : imaginary reward
2: Compute maximum reward in dataset $L := \max_{x \in \mathcal{D}} \tilde{r}_t(x)$
3: Set $U = R_{\max}$
4: **while** $U - L \geq \epsilon_1$ **do**
5:     Compute middle point $y_M = (U - L)/2$
6:     Perform conditional sampling $x_M \sim \widehat{P}(\cdot \mid y_M)$
7:     **if** $c(x_M) \leq \epsilon_2$ and $|\tilde{r}_t(x_M) - y_M| \leq \epsilon_3$ **then**
8:         Set $L = y_M$
9:     **else**
10:        Set $U = y_M$
11:    **end if**
12: **end while**
13: **Return** $x_t = x_M$

---

**Best-of-N sampling.** In practice, to improve the performances of both oracles presented above, it is possible to sample $N$ points $\mathcal{S}_N = \{x_t^1, \dots x_t^N\}$ via conditional generation, select the valid ones by checking $c(x_t^i) \leq \epsilon_c$ for a chosen value of $\epsilon_c$, and finally compute the maximum w.r.t. the imaginary reward $\tilde{r}_t$, namely $x_t := \arg\max_{x \in \mathcal{S}_N} \tilde{r}_t(x)$. This scheme is used by DIFFPS-$N$ in Sec. 6.

### F.2  PRACTICAL ALGORITHM IMPLEMENTATIONS

**Score Estimation and Sampling.** As already mentioned in 4, we don't train a conditional score at every iteration of the algorithm but leverage the fact that $\nabla_x \log p(x|y) = \nabla_x \log p(x) + \nabla_x \log p(x|y)$. We approximate $p(x|y) = \mathcal{N}(x^\top \theta, \sigma^2)$, with a fixed $\sigma$ and we approximate $\nabla_x \log p(x)$ using score matching. More formally, we use the following variance preserving SDE for the noise perturbation Song et al. (2020), the discretization of which corresponds to the forward diffusion in DDPM Ho et al. (2020).

$$\mathrm{d}x(k) = -\frac{1}{2}\beta(k)\mathrm{d}k + \sqrt{\beta(k)}\mathrm{d}w(k) \tag{24}$$

where $\beta(k) = \beta_{\min} + (\beta_{\max} - \beta_{\min})k$. As in Song et al. (2020), we choose $\beta_{\min} = 0.1$ and $\beta_{\max} = 20$. The objective that we minimize during training is the continuous weighted combination of fisher divergences that is given by:

$$\mathbb{E}_{k \sim \mathcal{U}(k_0, 1)}\left[\lambda(k)\mathbb{E}_{x(0) \sim p_0(x)}\mathbb{E}_{x(k) \sim p_k(\cdot \mid x(0))}[\|s(x(k), k) - \nabla_{x(k)} \log p_k(x(k)|x(k))\|]\right]$$

where:

$$p_k(x(k)|x(0)) = \mathcal{N}\left(e^{-\frac{1}{4}k^2(\beta_{\max}-\beta_{\min})-\frac{1}{2}k\beta_{\min}}x(0), I - Ie^{-\frac{1}{2}k^2(\beta_{\max}-\beta_{\min})-k\beta_{\min}}\right)$$

and we choose $\epsilon = 10^{-5}$ as well as $\lambda(k) = \sqrt{\mathbb{E}\|\nabla_{x(k)}\log p_k(x(k)|x(0))\|_2^2}$.

To solve the corresponding reverse SDE, we use a predictor corrector Song et al. (2020) and scale $\nabla_x \log p(x|y)$ by a factor $\gamma(t)$ that is decreasing in $k$ and hence the guidance strength is increased when solving the reverse SDE. We found this to be particularly useful in the case of linear rewards as in this setting, we cannot train a regressor/classifier on the noised samples, like one would typically do in guidance where the reward function is parameterized by a neural network. As $\tilde{r}_t$ is not invariant with respect to the projector $\Pi_V$ onto the manifold, we further use Tweedie's formula, to estimate the final sample one would obtain from unconditional sampling:

$$x_0 = \frac{x_k - (1 - \alpha_k)\nabla \log p_k(x_k)}{\sqrt{\alpha_k}}$$

where $\alpha_k = e^{-\frac{1}{2}t^2(\beta_{\max}-\beta_{\min})-k\beta_{\min}}$. We found that this allowed for effective guidance towards high reward regions. In the case of a linear reward function, we then use this estimate of $x_0$ in the conditional score $p(y|x_k) = \mathcal{N}(y; x_0^\top\theta, \sigma^2)$ and take the gradient w.r.t. $x_k$ meaning that we also differentiate through the estimated score.

### F.3 EXPERIMENTAL DETAILS

In the following section, we give further details on the implementation of DIFFPS in both experiments.

#### F.3.1 SPHERE ENVIRONMENT

**Data and Setup.** We consider the setting where $\Omega = \{x = Vz : \|z\|_1 \leq 1\}$ where $V \in \mathbb{R}^{D \times m}$ is a matrix that consists of the first $m$ columns of a matrix in the special orthogonal group, $SO(D)$. In order to generate the data, we sample $z$ uniformly from a unit sphere in $\mathbb{R}^m$ and then project it into $\mathbb{R}^D$. We choose $m = 4, D = 64$ and the number of samples $n = 1.2 \cdot 10^6$. Such high number of samples were necessary in order to be able to sample from high reward regions as outlined below.

**Reward and Cost.** As previously mentioned, we use a linear reward with a standard Gaussian prior on $\theta$ and the cost function is given as the L2 distance to the sphere in $D$ dimensions. Due to the fact that the reward maximum is always achieved at a single point on the surface of the sphere, we required a fairly large dataset, in order to be able to approximately sample those points.

**Neural Networks and Training Algorithms.** To parametrize the score function we use a 20-Layer MLP with skip connections and a hidden dimension of 128 neurons. For the time embedding we use Gaussian Random Features (Tancik et al., 2020). We train our model for 30 epochs with a batch size of 128, using the Adam optimizer with cosine annealing and warm restarts.

**Posterior Sampling.** We use the standard closed form updates for Gaussians given by (Russo et al., 2020):

$$\Sigma_{t+1} = \left(\Sigma_t + x_t x_t^\top / \sigma^2\right)^{-1}$$

$$\mu_{t+1} = \Sigma_{t+1}\left(\Sigma_t^{-1}\mu_t + x_t(y_t + \epsilon_t)/\sigma^2\right)^{-1}$$

where $(\mu_t, \Sigma_t)$ are the posterior mean and covariance, respectively and $\epsilon_t \sim \mathcal{N}(0, \sigma^2)$. We assume the noise $\sigma^2$ to be known and set it to 0.1. This also motivates the Gaussian likelihood $p(y|x)$ as explained in F.2.

**Best-of-N.** We set $N = 30$ and $\epsilon_c = 0.15$. If none of the 30 samples achieved a cost lower than this, we simply took the sample with the minimum cost. We also tried to the binary search oracle as presented in F.1 but found that the accuracy in the conditional generation required was too high, for the model we trained. In other words, we could not generate samples $x_M$ that achieved a reward close enough to $y_M$. We however believe that with an even better generative model, this method could be beneficial and could be explored further in the future.

