# OpenReview forum: "Generative bandit optimization via diffusion posterior sampling"
_ICLR.cc/2025/Conference — ICLR 2025 Conference Withdrawn Submission_

### Official Review · Reviewer_qPi9 · 2024-11-01

**Soundness:** 3
**Presentation:** 3
**Contribution:** 2
**Rating:** 5
**Confidence:** 3

**Summary:**

The paper introduces the Generative Bandit Optimization problem, aiming to solve high-dimensional discovery tasks like drug design by balancing exploration and exploitation in the search space. To achieve this, it proposes the Diffusion Posterior Sampling (DIFFPS) algorithm, which leverages conditional diffusion models to perform sampling directly on a learned data manifold, avoiding the curse of dimensionality. Theoretical analyses show DIFFPS's efficiency through bounds on Bayesian reward and misgeneration regret, adapting to the data's intrinsic dimensionality. Experimental results validate the algorithm's performance across different settings, supporting theoretical claims and demonstrating its robustness in high-dimensional spaces. This is beneficial to this community when the feature space is high.

**Strengths:**

1. The proposed DIFFPS algorithm is innovative in adapting diffusion models for bandit optimization, directly addressing the limitations of traditional bandit approaches in high-dimensional models in favor of the sparsity assumption.

2. The authors provide theoretical guarantees, demonstrating that DIFFPS’s performance aligns with the intrinsic dimensionality of the data. The bounds on Bayesian reward and misgeneration regret shows the tradeoff of the learned manifold and misspecificed action sets.

**Weaknesses:**

1. Despite its theoretical elegance, DIFFPS may present substantial implementation challenges due to the need for conditional diffusion models and unlabeled datasets. The reliance on substantial prior knowledge or offline data may limit its applicability in fully unsupervised or sparse data environments.

2. The paper lacks a detailed analysis of the algorithm’s sensitivity to hyperparameters, particularly the noise level, early stopping parameters (k0) and epsilon_c. This could impact DIFFPS's reliability across diverse datasets and real-world use cases, where optimal settings are difficult to determine.

3. While the experimental section compares DIFFPS with relevant baselines, broader comparison against a more diverse set of bandit and optimization algorithms, including recent model-based (high dimensional) approaches, could provide a fuller picture of DIFFPS's advantages and potential limitations.

4. Can this method be extend to the Bernoulli rewards and beta prior?

**Questions:**

1. What is the benefit of the different m and D. Say if m = 10%, 20%, 50%, 80% of D.

2. What is the novelty of the introducing of diffusion model and why introducing it? Use it to explore in the manifold? Is there any other sampling methods which can also work in the high dimensional study?

3. How to measure the misgeneration regret in real data? Why do we use the L2 distance as the candidate? Is there any other candidate?

4. What is the intuition in Algo line 5-7? Through learned P and y_{t}, sample the next candidate x_{t}, what is the intuition?

5. In theorem 5.1, why the reward regret ad mis-generation regret are in linear of T.

6. What is the computational complexity for the DIFFPS? Is it practical in real scenario? The comparison of computational time is missing.

---

### Official Review · Reviewer_nvSz · 2024-11-02

**Soundness:** 2
**Presentation:** 3
**Contribution:** 3
**Rating:** 5
**Confidence:** 2

**Summary:**

This paper proposes a generative bandit framework to describe applications where a generative model is used to sample instances with "high rewards" while navigating potentially high-dimensional but structured sample space with implicit constraints. With a set of unlabeled offline data, the authors propose an algorithm that progressively learns the conditional generative model for a high reward, samples an "action" (instance) from the model, obtain the reward, and update the generative model. Based on theory from the bandit literature, the authors prove the reward regret and constraint regret upper bounds under low-dimensional hidden space assumptions. The algorithm is empirically evaluated in a high-dimensional setting with linear rewards.

**Strengths:**

1. The problem setting is reasonable and captures useful application scenarios.
2. The generative bandit framework is a useful contribution that may provide the basis for future algorithmic & theoretical developments in the field.
3. The paper is comprehensive with framework, method, theory, and experiments.

**Weaknesses:**

1. The theory and experiments are a bit simplified thus disconnected from the motivating applications. While this can be acceptable in some cases, it also seems that the theory is mainly a combination of existing results. It will be helpful if technical contributions of such theory can be clarified. Otherwise, it might be more worthwhile to devote more efforts to stating the framework.

2. The statements on the contributions and distinctions from existing approaches can be improved. In particular, I was wondering where is the key contribution of this work when compared with the literature of guided diffusion models and black-box optimization, which also aim to generate samples that attain highest target values. Are they mainly in using the diffusion models and the theory in terms of intrinsic data dimension?

3. The clarity of introducing the framework can be improved. In particular, since the framework is quite complicated (involving low-d data manifold, diffusion model, conditional guiding, bandits, etc), readers may wonder where the essences of the framework are. Are there intuitions, which may relate to well-known results in diffusion models/bandits literature, for how the framework can gradually be geared towards feasible & high-value regions?

**Questions:**

Below are some additional questions.

1. In the experiments, how are the hyperparameters such as $k_0$ and $\nu$ chosen?

2. I'm not sure I understand how this framework tackles the unknown constraints. How does the lack of knowledge of the "feasible" region $\Omega$ come into theory? If it is learned from the offline dataset, what if $\hat\Omega$ cannot estimate $\Omega$ well?

3. How much does the linear reward setting captures the applicability of this framework to the motivating applications such as drug discovery?

---

### Official Review · Reviewer_wS1v · 2024-11-02

**Soundness:** 3
**Presentation:** 2
**Contribution:** 2
**Rating:** 5
**Confidence:** 3

**Summary:**

The paper introduces an approach for generative bandit optimization, focusing on complex discovery problems such as drug and material design where the solution space cannot be fully predefined. Classical bandit methods face challenges due to fixed action sets or high-dimensional continuous spaces, which lead to high complexities. To address these issues, the authors propose the Diffusion Posterior Sampling (DIFFPS) algorithm, which leverages conditional diffusion models to directly perform exploration and exploitation on a learned data manifold, which is an implicit, lower-dimensional representation of the action space. The paper includes theoretical guarantees demonstrating DIFFPS’s efficiency in adapting to this dimensionality and its capacity to minimize regret related to invalid actions.

**Strengths:**

The authors target high-dimensional bandit problems by employing diffusion models, which aim to optimize directly on a data manifold rather than across the entire action space. This method, DIFFPS, addresses some limitations of classical bandit algorithms in high-dimensional settings by adapting exploration to the intrinsic dimensionality of the data manifold, potentially reducing sample complexity.

**Weaknesses:**

1. The proposed algorithm requires multiple stages or processes, such as initializing with offline data, conducting conditional score matching, and performing posterior updates, but these stages are not clearly explained.

2. The authors make a number of assumptions, such as the low-dimensional linear subspace assumption for the data manifold, which may not hold or can be difficult to verify in various applications.

3. In numerical experiments, there is limited comparison with other generative models or alternative sampling methods that may also be capable of manifold learning.

**Questions:**

As mentioned in the part about weaknesses, does the proposed algorithm outperform other generative model-based bandit algorithms?

Please explain and give examples of real applications where the assumptions hold, and describe how to verify such assumptions concretely.

---

### Official Review · Reviewer_9R96 · 2024-11-04

**Soundness:** 2
**Presentation:** 3
**Contribution:** 2
**Rating:** 3
**Confidence:** 4

**Summary:**

This paper studies how to deal with high-dimensional bandit optimization problem. By formalizing the new generative bandit setting, an agent wishes to maximize an unknown reward function over the support of a data distribution, data manifold. And then diffusion posterior sampling is proposed to help take the action on the learned data manifold. Both theoretical and experimental results are provided.

**Strengths:**

1.	High-dimensional bandit optimization is a very important problem with many important applications in drug discovery, material design, and circuit design.
2.	On page 5, this paper shows discussion towards a practical and scalable algorithm due to practical considerations.
3.	Both theoretical and experimental results are provided to positively support the diffusion posterior sampling algorithm.
4.	Related work discussion is comprehensive and complete.

**Weaknesses:**

1.	My biggest concern lies in the novelty of this paper. As discussed in Section 7, DiffPS directly performs black-box function optimization on the approximate data manifold using a learned diffusion sampler, which in my opinion is very similar to work using dimension reduction first and then Bayesian optimization. Diffusion model is very popular these days, but the comparison of DiffPS and more existing work is heavily needed to show the unique advantage and motivation of DiffPS.
2.	I also have concerns on Assumption 5.1 and 5.2. By assuming low-dimensional linear subspace, Assumption 5.1 is too strong to model the real-world challenges in bandit optimization. Also, in Assumption 5.2, $VV^\top$ is linear and $\theta^\top_*$ is also linear, so $r_*(x)$ is a linear function of $x$?

**Questions:**

In Line 170 $x_t \in R^D$, in Figure 1 $x_t \in \Omega$, but in Line 297 $\Omega \subseteq R^m$?

---

### Note · Authors · 2024-11-26

**Comment:**

Eventually, we have decided to withdraw this paper towards strengthening it on the experimental side. We thank the reviewers for their observations.

**Withdrawal Confirmation:**

I have read and agree with the venue's withdrawal policy on behalf of myself and my co-authors.